# NPTX2 and cognitive dysfunction in Alzheimer's Disease

Mei-Fang Xiao[1,2,3†], Desheng Xu[1†], Michael T Craig[4], Kenneth A Pelkey[4], Chun-Che Chien[1], Yang Shi[1], Juhong Zhang[1], Susan Resnick[5], Olga Pletnikova[6], David Salmon[7,8], James Brewer[7,8], Steven Edland[8,9], Jerzy Wegiel[10], Benjamin Tycko[11], Alena Savonenko[6], Roger H Reeves[2,3], Juan C Troncoso[6,12], Chris J McBain[4], Douglas Galasko[7,8], Paul F Worley[1,12*]

[1]Solomon H. Snyder Department of Neuroscience, Johns Hopkins University School of Medicine, Baltimore, United States; [2]Department of Physiology, Johns Hopkins University School of Medicine, Baltimore, United States; [3]Institute for Genetic Medicine, Johns Hopkins University School of Medicine, Baltimore, United States; [4]Program in Developmental Neurobiology, Eunice Kennedy-Shriver National Institute of Child Health and Human Development, Bethesda, United States; [5]Laboratory of Behavioral Neuroscience, National Institute on Aging, Intramural Research Program, Baltimore, United States; [6]Department of Pathology, Johns Hopkins University School of Medicine, Baltimore, United States; [7]Department of Neurosciences, University of California San Diego Medical Center, San Diego, United States; [8]Shiley-Marcos Alzheimer's Disease Research Center, University of California San Diego Medical Center, San Diego, United States; [9]Division of Biostatistics and Bioinformatics, University of California San Diego, San Diego, United States; [10]Institute for Basic Research, New York City, United States; [11]Taub Institute for Research on Alzheimer's disease and the Aging Brain, Columbia University, New York City, United States; [12]Department of Neurology, Johns Hopkins University School of Medicine, Baltimore, United States

*For correspondence: pworley@jhmi.edu

†These authors contributed equally to this work

Competing interests: The authors declare that no competing interests exist.

**Abstract** Memory loss in Alzheimer's disease (AD) is attributed to pervasive weakening and loss of synapses. Here, we present findings supporting a special role for excitatory synapses connecting pyramidal neurons of the hippocampus and cortex with fast-spiking parvalbumin (PV) interneurons that control network excitability and rhythmicity. Excitatory synapses on PV interneurons are dependent on the AMPA receptor subunit GluA4, which is regulated by presynaptic expression of the synaptogenic immediate early gene NPTX2 by pyramidal neurons. In a mouse model of AD amyloidosis, *Nptx2*[-/-] results in reduced GluA4 expression, disrupted rhythmicity, and increased pyramidal neuron excitability. Postmortem human AD cortex shows profound reductions of NPTX2 and coordinate reductions of GluA4. NPTX2 in human CSF is reduced in subjects with AD and shows robust correlations with cognitive performance and hippocampal volume. These findings implicate failure of adaptive control of pyramidal neuron-PV circuits as a pathophysiological mechanism contributing to cognitive failure in AD.

## Introduction

The cause of memory loss in Alzheimer's disease remains an important unknown that negatively impacts therapeutics development (*De Strooper and Karran, 2016*). Amyloid Aß that accumulates in human brain reduces excitatory synaptic function (*Shankar et al., 2008*) and mouse models of Aß

amyloidosis document that Aß causes weakening and loss of excitatory synapses (*Kamenetz et al., 2003*; *Um et al., 2013*). Aß also increases pyramidal neuron excitability consequent to down-regulation of voltage sensitive ion channel Nav1.1 in PV inhibitory interneurons (*Verret et al., 2012*). However, human studies indicate amyloid accumulation typically precedes cognitive failure by many years (*Jack and Holtzman, 2013*). Moreover, case studies report patients who were cognitively normal at death yet show pronounced Aß plaque and neurofibrillary tangles consistent with AD (*Driscoll and Troncoso, 2011*). Mouse models of Aß amyloidosis show cognitive improvement following inhibition of Aß generating enzymes or amyloid reducing therapies (*Comery et al., 2005*; *Kimura et al., 2010*; *Townsend et al., 2010*), yet similar successes have not been reported in human trials. This discrepancy highlights the need for expanded understanding of cognitive failure in human AD.

We focused attention on carefully archived human brain and cerebrospinal fluid samples to assess mechanisms important for cognitive function in AD. Extending from observations of altered brain activity and its possible contribution to human AD progression (*Buckner et al., 2009*, *Buckner et al., 2005*) we assayed proteins of the class of cellular immediate early genes (IEGs), which mediate activity-dependent plasticity important for memory function and can be assayed in human post-mortem tissue (*Wu et al., 2011*). Here, we report that Neuronal Pentraxin 2 (NPTX2; also termed Narp and NP2) is down-regulated in brain of human subjects with Alzheimer's disease (AD). NPTX2 is a member of a family of 'long' Neuronal Pentraxins that includes Neuronal Pentraxin 1 (NPTX1) and Neuronal Pentraxin Receptor (NPTXR). NPTX1, NPTX2 and NPTXR form disulfide-linked, mixed NPTX complexes that traffic to the extracellular surface at excitatory synapses (*Xu et al., 2003*) where they bind AMPA type glutamate receptors and contribute to multiple forms of developmental and adult synaptic plasticity (*Chang et al., 2010*; *Cho et al., 2008*; *Gu et al., 2013*; *Lee et al., 2017*; *O'Brien et al., 2002*, *1999*; *Pelkey et al., 2015*, *2016*). NPTXs directly bind AMPA type glutamate receptors and can act in 'trans' as presynaptic factors that induce postsynaptic excitatory synapses (*Lee et al., 2017*; *O'Brien et al., 2002*, *1999*; *Sia et al., 2007*). NPTX2 expressed by pyramidal neurons and secreted by axon terminals uniquely mediates activity-dependent strengthening of pyramidal neuron excitatory synapses on GABAergic PV-interneurons (*Chang et al., 2010*). This adaptive strengthening of the pyramidal neuron-PV interneuron synapse mediates homeostatic scaling of the circuit, a process implicated in the ability of circuits to encode information (*Turrigiano, 2012*). We examined the consequence of NPTX2 reduction in a mouse model of AD and found that amyloidosis together with $Nptx2^{-/-}$ results in a synergistic reduction of inhibitory circuit function in conjunction with a reduction of the AMPA type glutamate receptor GluA4. GluA4 is preferentially expressed in PV-interneurons at excitatory synapses where it co-localized with NPTX2 (presynaptic source) (*Chang et al., 2010*) and confers rapid channel inactivation (*Angulo et al., 1997*; *Geiger et al., 1995*) that is essential for fast spiking and contributions of PV interneurons to hippocampal circuit rhythmicity and memory (*Fuchs et al., 2007*). $Nptx2^{-/-}$ dependent reduction of GluA4 was previously described in a different mouse model ($Nptx2^{-/-}$; $Nptxr^{-/-}$) where analysis suggested NTPX2 is required to bind and stabilize postsynaptic expression of GluA4 in PV-interneurons in conditions of altered activity (*Pelkey et al., 2015*, *2016*). In human brain GluA4 is also selectively expressed in PV-INs and is reduced in AD brain regions where levels correlate within samples to NPTX2. This suggests that NPTX2 reduction in human AD brain is linked to disruption of pyramidal neuron-PV interneuron circuits. To assess the relation between NPTX2 and cognitive function in human subjects, we determined that NPTX2 is present in CSF and levels are representative of brain NPTX2 since CSF NPTX2 discriminates clinically defined AD compared to age-matched controls. Importantly, NPTX2 levels in AD subjects correlated with cognitive performance employing comprehensive psychometric tests including the Dementia Rating Scale, and surpassed the performance of other CSF markers including Aß42, tau and p-tau. NPTX2 levels also correlated with MRI measures of hippocampal volume, which is considered a marker of neurodegeneration and is linked to cognitive decline (*Jack et al., 2010*; *Weiner et al., 2015*). NPTX2 is not down-regulated in a widely used mouse amyloidosis model and is distinct from other CSF markers attributed to neurodegeneration including tau and p-tau in that NPTX2 is an indicator of specific rather than general excitatory synapses. Studies support the hypothesis that NPTX2 down-regulation results in disruption of pyramidal neuron-PV interneuron circuits that are important for brain rhythmicity and homeostasis of excitability, and represents a previously unrecognized mechanism important for human cognitive dysfunction and progression in AD.

## Results

### NPTX2 is reduced in human AD and Down syndrome brain

NPTX2 protein was assayed by western blot (WB) in human brain from individuals with pathologically confirmed late onset Alzheimer's disease (AD) versus age-matched controls (*Figure 1A and B*, *Figure 1—figure supplement 1* and *Figure 1—source data 1*). NPTX2 was reduced in all cortical regions including those areas that display prominent classical neuropathological changes in AD, as well as areas that are typically less affected such as occipital cortex. Hippocampus was not available for many cases and could not be systematically examined. Reductions of NPTX2 were evident whether referenced to actin (*Figure 1—figure supplement 1A*) or PSD95 (*Figure 1B*); the latter indicating that NPTX2 down-regulation is distinct from a general reduction of excitatory synaptic markers. *Nptx2* mRNA was also prominently reduced in assayed regions (*Figure 1C*). Neuronal Pentraxin 1 (NPTX1) and Neuronal Pentraxin Receptor (NPTXR) were not reduced in the same brain samples (*Figure 1D and E*). To further assess the specificity of NPTX2 down-regulation we determined that other IEGs including Arc and Egr-1 were not reduced (*Figure 1—figure supplement 1B*). NPTX2 was not reduced in brain of subjects who were cognitively normal at death but whose brains exhibit pathology typical of AD including Aß plaque and tangles [asymptomatic AD (ASYMAD) or preAD (*Codispoti et al., 2012*; *Driscoll et al., 2006*; *Driscoll and Troncoso, 2011*) (*Figure 1F and G*, and *Figure 1—source data 1*). *Nptx2* protein and mRNA were also reduced in middle frontal gyrus of Down syndrome (DS) subjects aged 19 y/o to 40 y/o compared to age matched controls (*Figure 1H* to 1J and *Figure 1—source data 2*). DS individuals exhibit a high risk for AD after the age of 40 (*Zigman and Lott, 2007*). *Nptx2* protein (normalized to actin) and mRNA were similarly reduced in older DS individuals with AD (*Figure 1—figure supplement 2* and *Figure 1—source data 3*).

### A microRNA mechanism of NPTX2 down-regulation in human AD brain

We examined determinants of *Nptx2* mRNA down-regulation in human brain. *Nptx2* transcription is regulated by methylation of flanking genomic sequences in pancreatic cells (*Zhang et al., 2012*), however, *Nptx2* methylation assayed by pyrosequencing was low in human brain and not different between control and AD subjects (*Figure 2—figure supplement 1*). Furthermore, *Nptx2* pre-mRNA expression was not different between control and AD subjects (*Figure 2A*) suggesting *Nptx2* transcription is maintained in AD, and that reduced mRNA is consequent to reduced pre-mRNA processing (ex. splicing) or mRNA stability. IEGs are targets of miRNA control (*Huang et al., 2012*) and TargetScan predicted several candidate miRNAs that target *Nptx2* 3' UTR (*Figure 2B*). We determined that *miR-1271* as well as its originating pri-miR, pre-mRNA and mRNA [*miR-1271* is generated from intron of human gene *Arl10* (ADP-ribosylation factor-like 10)] were increased in AD frontal cortex where levels in individual samples correlated with *Nptx2* mRNA (*Figure 2C*, and *Figure 2—figure supplement 2A*). Analysis of AD precuneus gyrus detected similar preservation of *Nptx2* pre-mRNA (*Figure 2—figure supplement 2B*), and increases of *pri-miR-1271*, *Arl10* pre-mRNA and mRNA, as well as increases of *miR-182* (*Figure 2—figure supplement 2C*). For comparison, we examined frontal cortex of DS cases and confirmed up-regulation of the triploid *miR-155* (positive control) together with increases in *miR-96* and *miR-182* (*Figure 2D*). *miR-96*, *miR-182* and *miR-1271* target the same sequence in *Nptx2* 3'UTR (*Figure 2E*). We confirmed in heterologous cells that all three miRs reduce expression of a luciferase fusion encoding human *Nptx2* 3'UTR compared to a point mutant 3' UTR that prevents miR targeting (*Figure 2—figure supplement 2E–F*). miRs expressed as pri-miRs by lentivirus in cultured mouse cortical neurons reduced native mouse NPTX2 protein expression (*Figure 2F* to 2 H). *miR-96* was most effective and additionally reduced *Nptx2* mRNA but not pre-mRNA. Caveats for these experiments include the artificially high level of lentiviral miR expression, possible species differences in the response, and the acuteness of the manipulation relative to human disease. Nevertheless, these studies indicate that NPTX2 down-regulation in human AD and DS brain occurs consequent to dysregulation of mRNA after transcription and suggest a role for miR mechanisms.

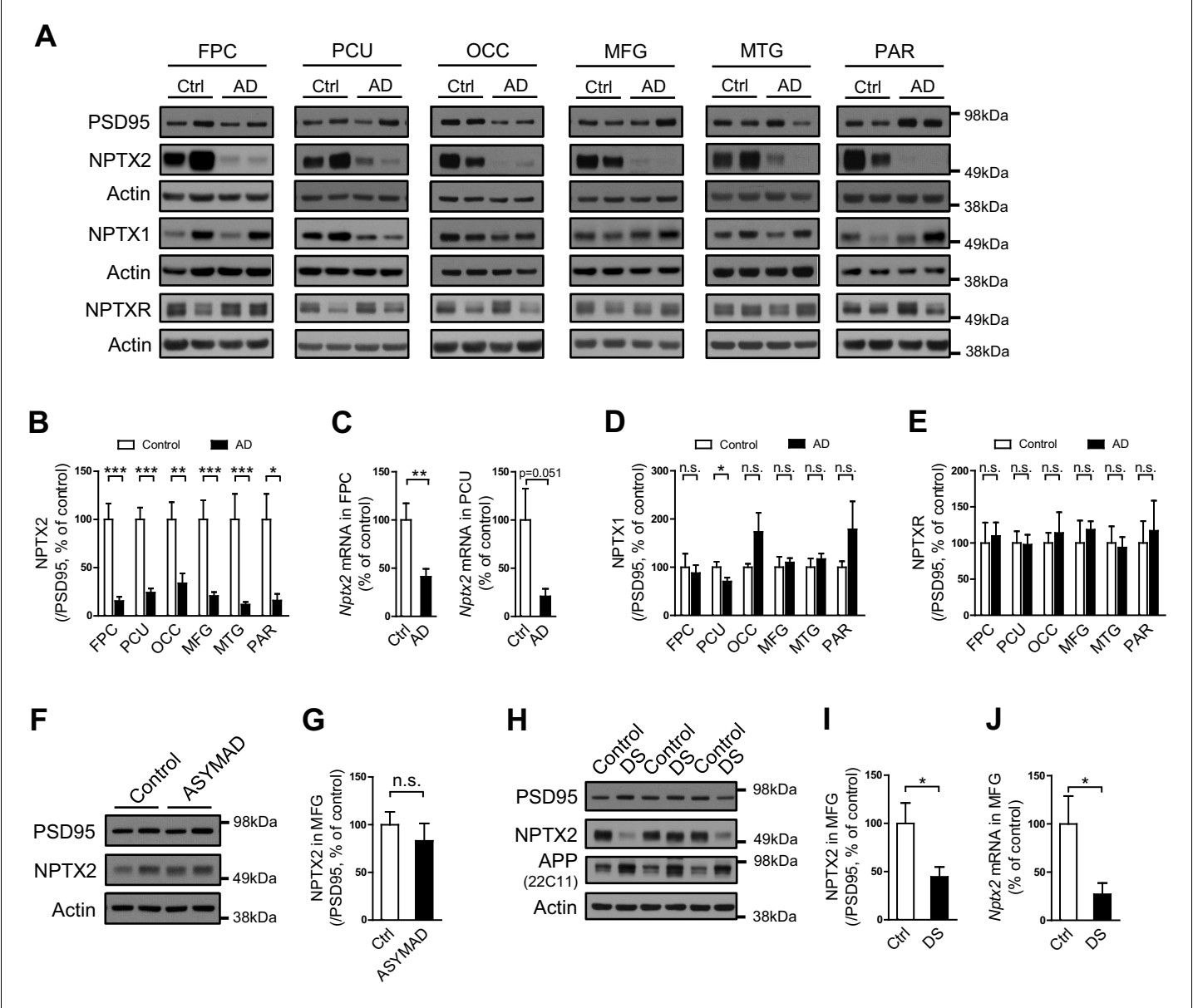

**Figure 1.** NPTX2 levels are reduced in human postmortem AD brain and DS brain, but not in ASYMAD brain. (**A,B,D** and **E**) Representative western blot images (**A**) and quantification of NPTX2 (**B**), NPTX1 (**D**) and NPTXR (**E**) normalized to PSD95 in the frontopolar cortex (FPC), precuneus (PCU), occipital gyrus (OCC), middle frontal gyrus (MFG), middle temporal gyrus (MTG) and parietal gyrus (PAR) from controls and AD subjects. NPTX2 is down-regulated in all assayed brain regions of AD individuals. FPC: control, n = 7; AD, n = 8. PCU: control, n = 15; AD, n = 19. OCC: control, n = 7; AD, n = 7. MFG: control, n = 9; AD, n = 16. MTG: control, n = 10; AD, n = 18. PAR: control, n = 5; AD, n = 5. (**C**) *Nptx2* mRNA is reduced in AD brain. FPC: control, n = 9; AD, n = 16. PCU: control, n = 7; AD, n = 6. (**F,G**) Western blot assays reveal no significant change of NPTX2 expression in MFG from subjects with asymptomatic AD (ASYMAD). Control, n = 8; ASYMAD, n = 10. (**H, I**) Western blot assays show significant reduction of NPTX2 in MFG of individuals with Down syndrome (DS). Control, n = 6; DS, n = 6. (**J**) *Nptx2* mRNA is reduced in MFG of individuals with DS. Control, n = 5; DS, n = 5. *p<0.05, **p<0.01, ***p<0.001 by two-tailed t test. Data represent mean ± SEM.

The following source data and figure supplements are available for figure 1:

**Source data 1.** Clinical and histopathological information of ASYMAD and AD individuals for brain analysis.
**Source data 2.** Information of individuals with Down syndrome for brain analysis.
**Source data 3.** Information of individuals with Down syndrome and Alzheimer's disease for brain analysis.

*Figure 1 continued*

**Figure supplement 1.** Immediate early gene expression in human postmortem AD brain.

**Figure supplement 2.** NPTX2 levels are reduced in human postmortem DS-AD brain.

### *Nptx2* ko and Aß amyloidosis synergistically disrupt hippocampal rhythmicity

NPTX2 is not reduced in brain of 6 month old *APPswe/PS1ΔE9* mice (here termed *hAPP*) (*Borchelt et al., 1997*) (*Figure 3—figure supplement 1*). To assess the possible impact of NPTX2 down-regulation in human AD brain, we created a mouse model that combines amyloidosis (*APPswe/PS1ΔE9*; here termed *hAPP*) and *Nptx2$^{-/-}$*. Amyloidosis in a similar mouse model increases hippocampal excitability by reducing excitability of PV-interneurons (*Palop et al., 2007*; *Verret et al., 2012*). Accordingly, we considered the possibility that loss of NPTX2 would further reduce PV neuron function. We prepared acute hippocampal slices from 3–4 month old mice and recorded extracellular field potentials from stratum pyramidale of CA3 from WT, *Nptx2$^{-/-}$*, *hAPP* and *hAPP/Nptx2$^{-/-}$*. Spontaneous sharp wave ripples (SWR) were evident in all four groups (*Figure 3—figure supplement 2A–2D*). SWR are spontaneous oscillatory currents that are dependent on PV-interneuron function (*Csicsvari et al., 1999*; *Ellender et al., 2010*) and reflect the near synchronous firing of pyramidal neuron ensembles. SWRs occurred at a significantly higher incidence, but with a lower peak frequency, in *hAPP/Nptx2$^{-/-}$* mice (*Figure 3—figure supplement 2E–2F*). We next monitored gamma oscillations induced by bath-application of carbachol (*Fisahn et al., 1998*). Gamma oscillations are field potentials recorded near the soma of pyramidal neurons and reflect the synchronized inhibitory currents created by PV interneuron synaptic drive (*Buzsáki and Wang, 2012*). In both WT and *Nptx2$^{-/-}$* mice, gamma oscillations were observed with a peak frequency of 30 to 40 Hz (*Figure 3A and B*). *hAPP* mice displayed a regular gamma-like rhythm, but at a lower frequency (*Figure 3C*). *hAPP/Nptx2$^{-/-}$* exhibited even slower 'gamma' oscillation with significantly reduced peak power (*Figure 3D* to 3F). Additionally, rhythmicity was disrupted by prominent hypersynchronous bursts (*Figure 3D*) indicative of increased network excitability.

### *Nptx2* ko and Aß amyloidosis synergistically disrupt hippocampal GluA4 expression

We sought evidence for a specific role for *Nptx2$^{-/-}$* on pyramidal neuron-PV interneuron function in *hAPP/Nptx2$^{-/-}$* mice. NPTX2 is required for homeostatic up regulation of GluA4 at excitatory synapses on PV interneurons in response to increased pyramidal neuron activity (*Chang et al., 2010*). GluA4 expression is not substantially altered in brain of *NPTX2$^{-/-}$* mice (*Figure 3G*), however, GluA4 is markedly reduced in *Nptx2$^{-/-}$/Nptxr$^{-/-}$* mice (*Pelkey et al., 2015*, *2016*). Since NPTXR is required for mGluR-LTD (*Cho et al., 2008*), *Nptxr$^{-/-}$* presumably increases pyramidal neuron excitability and demand for NPTX2 to bind and stabilize GluA4. We monitored GluA4 expression and found a marked reduction in *hAPP/Nptx2$^{-/-}$* compared to WT control or *hAPP* (*Figure 3H*). Neither *Nptx2$^{-/-}$* alone (*Figure 3G*) (*Pelkey et al., 2015*, *2016*) nor *hAPP* alone (*Figure 3H*) reduced GluA4 expression indicating that GluA4 reduction in *hAPP/Nptx2$^{-/-}$* represents an emergent phenotype that impacts PV interneuron function. While there are important limitations of *hAPP/Nptx2$^{-/-}$* mice as a model of AD including supra pathophysiological levels of Aß and complete deletion of NPTX2, these studies identify GluA4 expression and effects on inhibitory circuit function as consequences of the combined action of amyloidosis and NPTX2 down-regulation.

### GluA4 expression correlates with NPTX2 in both aged human control and AD subjects

As in mouse brain, GluA4 expression is selectively enriched in PV-interneurons in human brain (*Figure 4A*). We examined GluA4 expression in multiple brain regions in both AD subjects and age-matched controls. GluA4 expression was reduced in AD brain regions including precuneus and medial frontal gyri (*Figure 4B and C*). The precuneus gyrus (Brodmann Area 7) resides in the parietal cortex and is part of the 'default network' with strong connections to the hippocampus

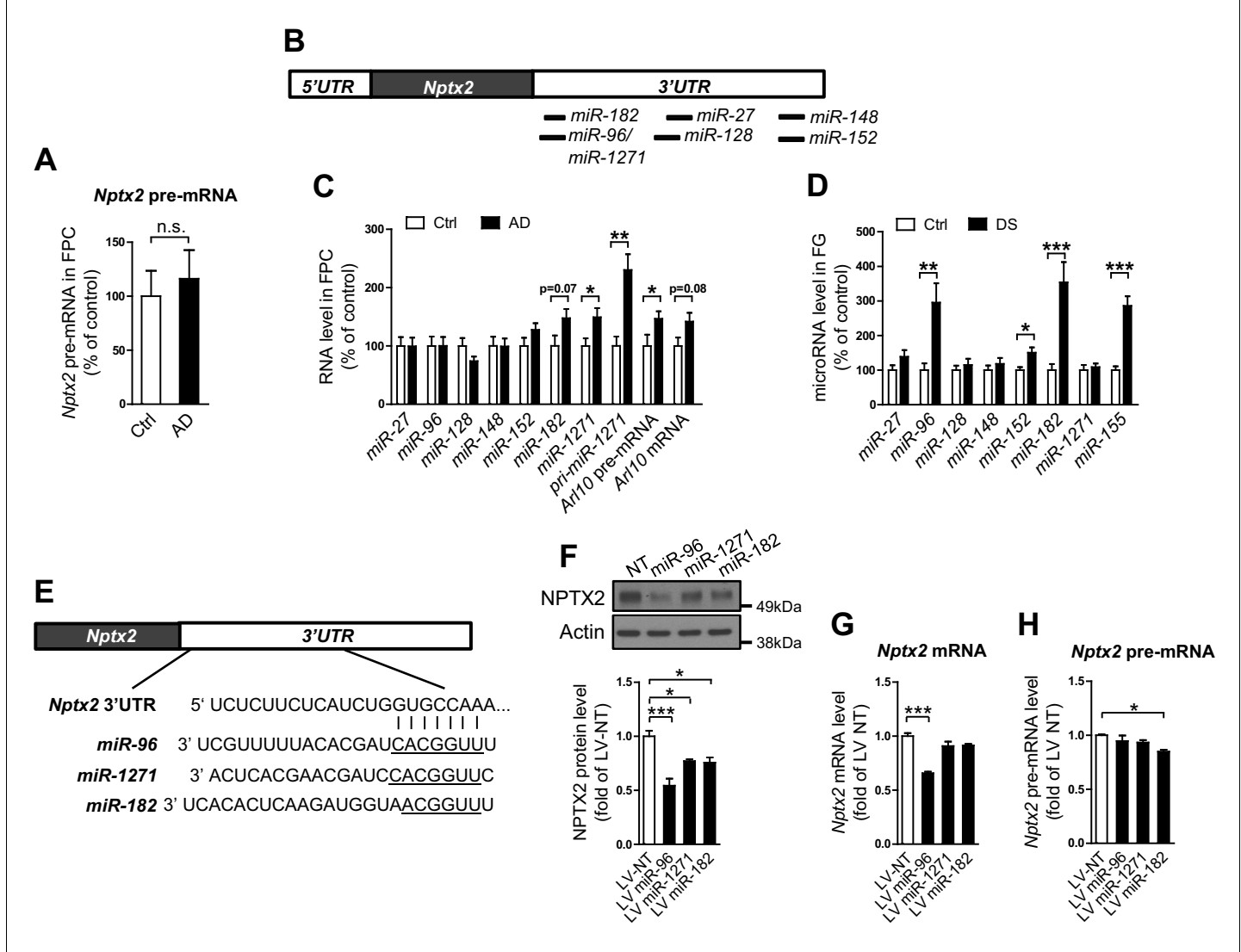

**Figure 2.** miRNAs dysregulation and NPTX2 down-regulation in AD brains. (**A**) *Nptx2* pre-mRNA levels are identical in frontopolar cortex (FPC) of AD subjects and control. Control, n = 9; AD, n = 16. (**B**) microRNAs predicted to bind with *Nptx2* 3'UTR by TargetScan. (**C**) Taqman assays show *miR-182* and *miR-1271* are increased in FPC of AD individuals. Control, n = 9; AD, n = 16. *p<0.05, **p<0.01 by two-tailed t test. (**D**) *miR-96, miR-152* and *miR-182* are up-regulated in individuals with Down syndrome (DS). Triploid *miR-155* served as positive control. Control, n = 14; DS, n = 18. *p<0.05, **p<0.01, ***p<0.001 by two-tailed t test. (**E**) *miR-96, miR-182* and *miR-1271* target the same sequence in *Nptx2* 3'UTR. (**F–H**) Cultured mouse cortical neurons are transduced with lentivirus expressing nontargeting miRNA (LV-NT) or *miR-96, miR-1271* and *miR-182*. (**F**) Expression of *miR-96, miR-1271* and *miR-182* reduce NPTX2 protein level. (**G**) miR-96 reduces *Nptx2* mRNA. (**H**) *Nptx2* pre-mRNA is preserved by *miR-96* and *miR-1271* expression. n = 5 wells from three independent culture except n = 4 wells for LV-NT group in *Figure 2G*. *p<0.05, ***p<0.001 by nonparametric one way ANOVA with Tukey post hoc test. Data represent mean ± SEM.

The following figure supplements are available for figure 2:

**Figure supplement 1.** Methylation of *Nptx2* promoter in human brain.

**Figure supplement 2.** *Nptx2* mRNA is targeted by miRNAs that are upregulated in AD brain.

(*Buckner et al., 2008*; *Greicius et al., 2004*; *Vincent et al., 2006*) and is notable for high levels of amyloid detected by PET in AD subjects and some aged control individuals (*Sperling et al., 2009*). The histopathology of the precuneus gyrus in AD is not different from surrounding brain regions (*Nelson et al., 2009*). Interesting, when GluA4 expression was evaluated separately in controls and

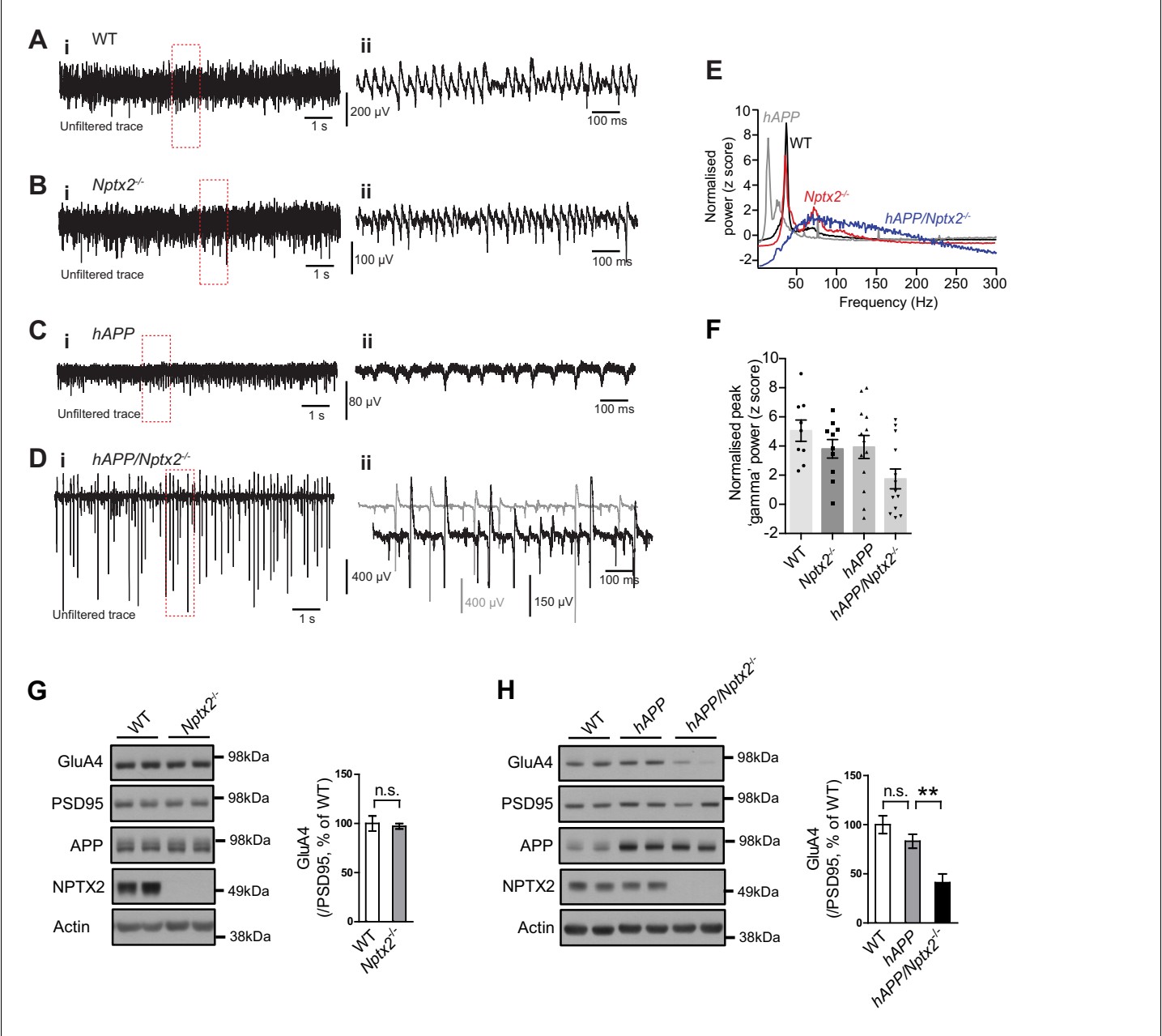

**Figure 3.** Circuit Rhythmicity and GluA4 expression are disrupted in *hAPP/Nptx2⁻/⁻* mice. (A–D) Example extracellular field potentials (i) with hatched area shown on an expanded time base (ii) for WT (A), *Nptx2⁻/⁻* (B), *hAPP* (C), and *hAPP/Nptx2⁻/⁻* (D) mice. For *hAPP/Nptx2⁻/⁻* trace, grey trace in (ii) shows trace on expanded time base and the black trace shows expanded voltage axis to show gamma-like oscillations nested within population spikes. (E) Power spectra for WT, *Nptx2⁻/⁻*, *hAPP*, and *hAPP/Nptx2⁻/⁻* mice, taken from 400 s of recording, normalised to the power between 3 and 300 Hz. (F) The normalised power of gamma-like oscillations was significantly reduced in *hAPP/Nptx2⁻/⁻* mice (z-score; WT, 5.1 ± 0.73; *Nptx2⁻/⁻*, 3.8 ± 0.63; *hAPP*, 3.9 ± 0.79; *hAPP/Nptx2⁻/⁻*, 1.7 ± 0.68; p=0.0199, One-way ANOVA). **, p<0.01 *vs* WT, *post-hoc* multiple comparisons test. (G) GluA4 expression is not altered in 6 month-old *Nptx2⁻/⁻* mouse cortex. n = 4 for *WT* and n = 3 for *Nptx2⁻/⁻*. (H) Representative western blot images and quantification of GluA4 in forebrains of 6 month-old WT, *hAPP* and *hAPP/Nptx2⁻/⁻* mice. *hAPP/Nptx2⁻/⁻* mice show reduced GluA4 in brain. WT, n = 4; *hAPP,* n = 6; *hAPP/ Nptx2⁻/⁻*, n = 4. **p<0.01 by nonparametric one way ANOVA with Tukey post hoc test.

The following figure supplements are available for figure 3:

**Figure supplement 1.** NPTX2 expression in amyloidosis mouse model.

**Figure supplement 2.** Spontaneous sharp-wave ripples (SWRs) occur more frequently in *hAPP/Nptx2⁻/⁻* mice.

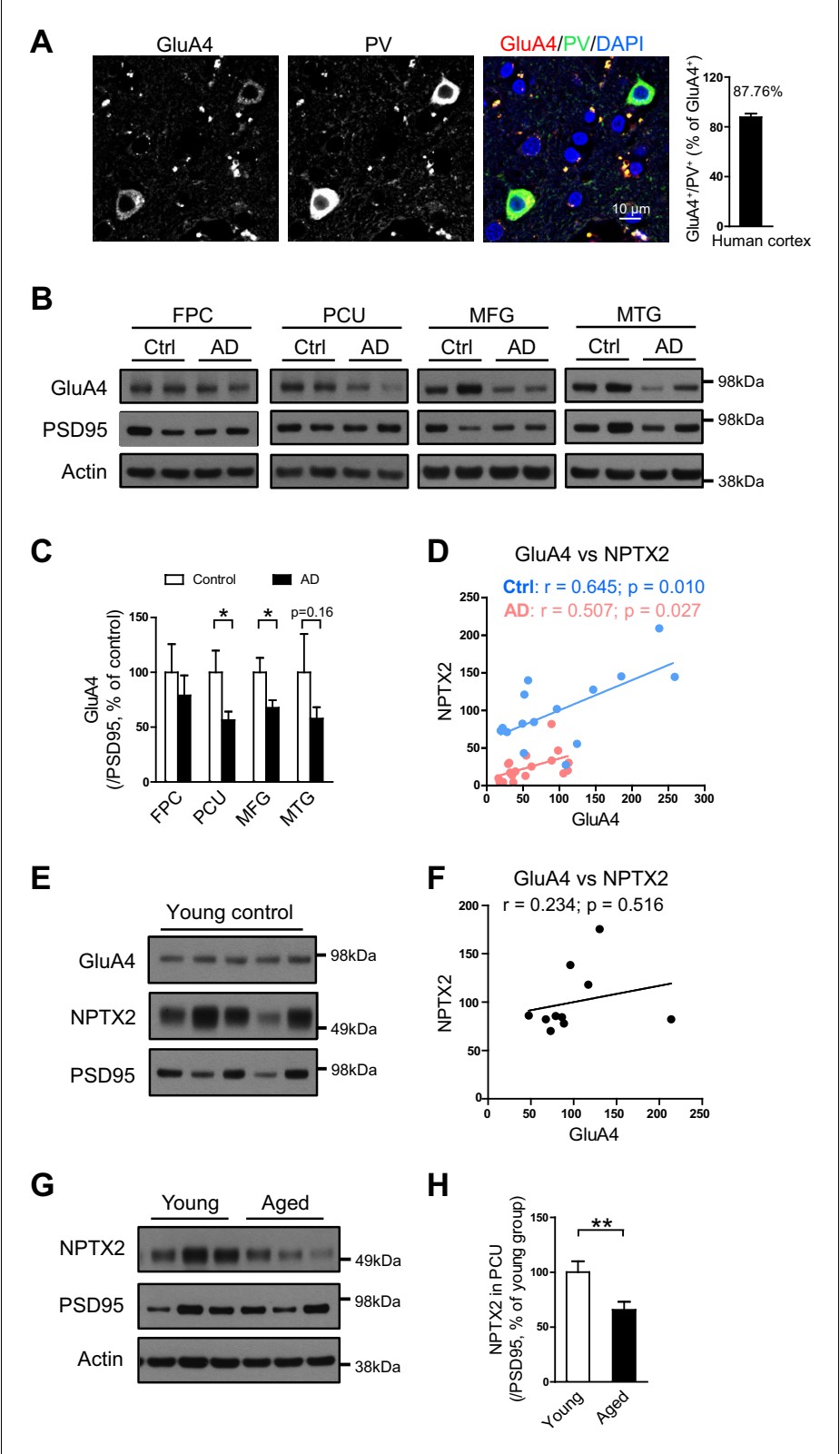

**Figure 4.** GluA4 levels are reduced in human postmortem AD brain. (**A**) Immunostaining of GluA4 demonstrates GluA4 is enriched on PV-IN in human cortex. Data were collected from four cases including occipital gyrus and parietal gyrus. (**B** and **C**) Representative western blot images (**B**) and quantification of GluA4 (**C**) in the frontopolar cortex (FPC), precuneus (PCU), middle frontal gyrus (MFG) and middle temporal gyrus (MTG) from controls and AD subjects. GluA4 is significantly down-regulated in PCU and MFG of AD individuals. FPC: control, n = 7; AD, n = 8. PCU: control, n = 15;
*Figure 4 continued on next page*

*Figure 4 continued*

AD, n = 19. MFG: control, n = 9; AD, n = 16. MTG: control, n = 10; AD, n = 18. *p<0.05 by two-tailed t test. Data represent mean ± SEM. (D) GluA4 levels correlate with NPTX2 in both control and AD group. Pearson correlation coefficient analysis was performed. (E and F) GluA4 expression did not correlate with NPTX2 in young adult brain. n = 10. Pearson correlation coefficient analysis was performed. (G and H) NPTX2 expression in young adult brain was higher than in aged controls. Young, n = 12; Aged, n = 15. **p<0.01 by two-tailed t test. Data represent mean ± SEM.

The following source data is available for figure 4:

**Source data 1.** Information of young healthy controls and aged healthy controls for brain analysis.

AD subjects there was a strong correlation with NPTX2 expression in both groups, albeit levels of NPTX2 and GluA4 were lower in AD (*Figure 4D*). In young adult brain GluA4 expression did not correlate with NPTX2 (*Figure 4E and F*), and NPTX2 expression was higher than in aged controls (*Figure 4G and H*, and *Figure 4—source data 1*). These observations suggest that NPTX2 expression is reduced with normal aging and becomes a determinant of GluA4 expression independent of overt cognitive disease. GluA4 is further reduced in AD by NPTX2 down-regulation in the context of amyloidosis.

## CSF NPTX2 provides a biomarker of brain NPTX2

To assess the relationship of NPTX2 to cognitive performance, we needed to establish a bioassay of brain NPTX2 that could be measured in living subjects for whom behavioral test scores and other information were available. We detected NPTX2 in CSF by WB, consistent with its natural secretion at excitatory synapses and its potentially reversible association with glycoprotein networks that uniquely surround PV-interneurons (*Chang et al., 2010*). CSF NPTX2 levels were reduced in AD subjects (*Figure 5A and B*, and *Figure 5—source data 1*). CSF NPTX1 and NPTXR levels were also reduced in the same CSF samples (*Figure 5A and B*, and *Figure 5—figure supplement 1*). NPTXR in CSF was detected in WB as three bands corresponding to full length (60 kDa) and presumed cleavage products at 45 kDa and 30 kDa and all three were reduced in AD CSF (*Figure 5—figure supplement 1*). NPTX1 and NPTXR are widely expressed at excitatory synapses (*Lee et al., 2017*; *Xu et al., 2003*) and their reduction in CSF despite preserved levels in AD brain suggests that their expression in CSF arises from a discrete source. We determined that NPTX1 and NPTX2 in human CSF are disulfide-linked oligomers (*Figure 5—figure supplement 2A*) consistent with their known co-assembly in brain (*Xu et al., 2003*). Moreover, levels of CSF NPTX1 and NPTXR strongly correlate with CSF NPTX2 in individual samples (*Figure 5—figure supplement 2B*). Since NPTX2 selectively accumulates on PV-interneurons, we infer this as the major source of NPTX1, NPTX2, and NPTXR in CSF.

We developed a sandwich ELISA to quantitate NPTX2 protein in CSF (*Figure 5—figure supplement 3*) and confirmed close correspondence with levels measured by WB (*Figure 5C*). The mean NPTX2 level in control CSF was 1067 pg/ml compared to 296 pg/ml in AD (*Figure 5D*). Consistent with current standards to document a new biomarker (*Noel-Storr et al., 2014*) we screened a second independent set of patient CSF samples by WB and ELISA and confirmed consistency of levels in controls and reduction of NPTX2 in AD (*Figure 5E*, *Figure 5—figure supplement 4*, and *Figure 5—source data 2*). NPTX2 ELISA of samples from patients with MCI demonstrated reduction of NPTX2 compared to controls (*Figure 5F*).

We compared the performance of NPTX2 to standard CSF biomarkers of AD. Receiver operating characteristic (ROC) analysis revealed the specificity, sensitivity and accuracy of NPTX2 was similar to Aß42 and superior to tau and p-tau for distinguishing controls from AD (*Figure 5G–J*). A ratio of tau or p-tau divided by NPTX2 showed improved performance (*Figure 5K and L*). NPTX1/2/R levels did not correlate with Aß42 but did positively correlate with tau and p-tau within individual AD patients (*Figure 5—figure supplement 5*) and controls (NPTX2 vs tau, r = 0.4625, p=0.0002; NPTX2 vs p-tau, r = 0.3607, p=0.0043, Pearson correlation coefficient analysis). This suggests that tau appearance in CSF may be linked to NPTX2 and PV circuit function, however since they change in opposite directions in AD and yet are positively correlated in both controls and AD their association appears complex. Moreover, their complementarity in diagnostic performance suggests they detect distinct processes.

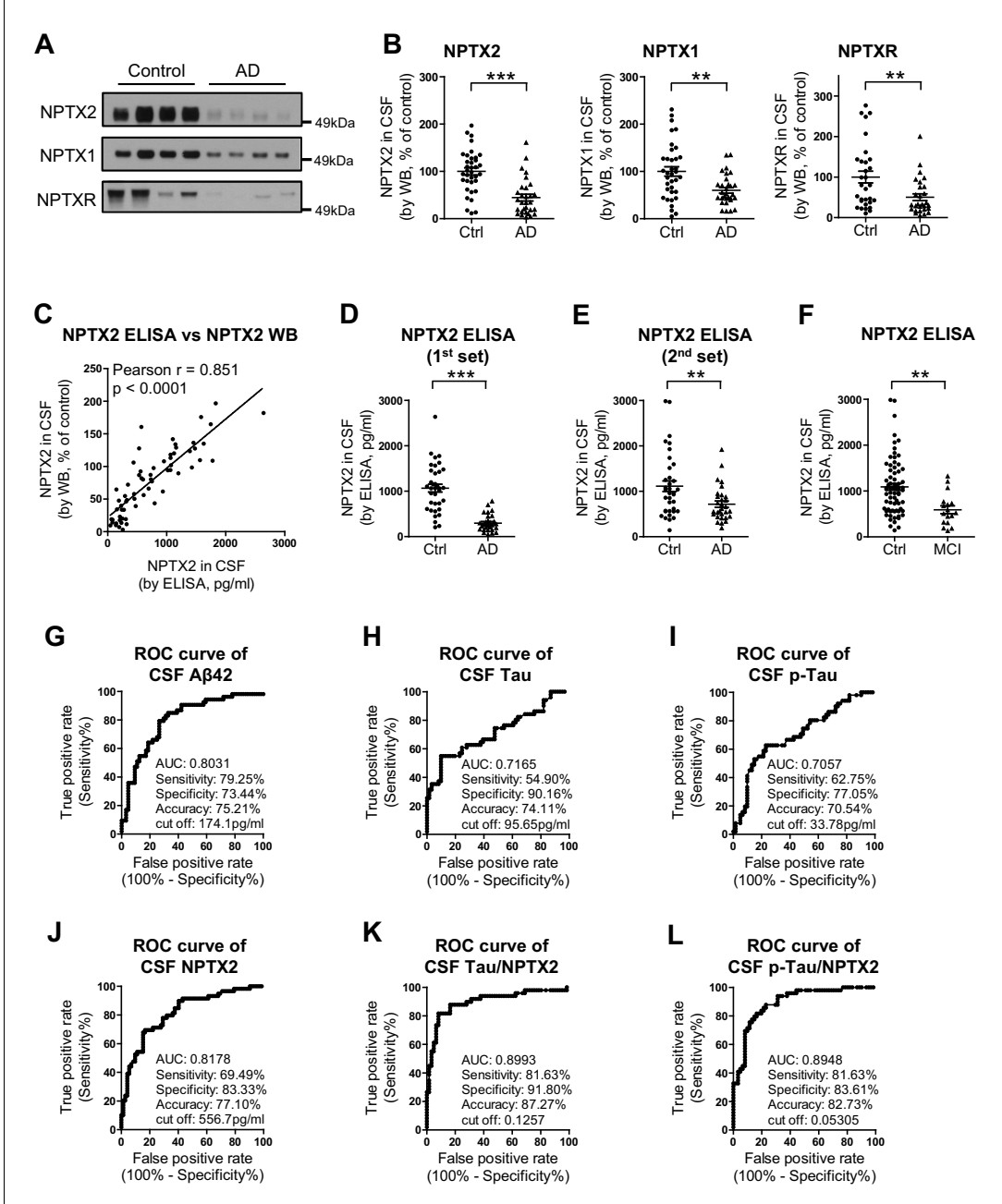

**Figure 5.** NPTX levels are reduced in CSF from individuals with clinically diagnosed AD. (A,B) Western blot assays of NPTX2, NPTX1 and NPTXR in lumbar cerebrospinal fluid (CSF) from age-matched controls and patients with clinically diagnosed AD. AD patients show reduced NPTX2, NPTX1 and NPTXR levels in CSF. Control, n = 36; AD, n = 30. (C) ELISA was developed to quantitate NPTX2 protein in CSF. NPTX2 assayed by ELISA corresponds closely with levels defined by WB. n = 64. Pearson correlation coefficient analysis was performed. (D) ELISA shows significant reduction of NPTX2 in CSF from patients with clinically diagnosed AD. Control, n = 36; AD, n = 28. (E) ELISA assay confirmed the reduction of NPTX2 in AD in second set of CSF sample. n = 36 for control, n = 30 for AD. (F) CSF NPTX2 levels are significantly reduced in individuals with mild cognitive impairment (MCI) compared with healthy controls. Control, n = 72; MCI, n = 17. (G–L) Receiver operating characteristic (ROC) curve analysis of CSF Aβ42 (G), tau (H), p-tau181 (I), NPTX2 (J), tau/NPTX2 (K) and p-tau/NPTX2 (L) as AD diagnostic biomarkers for distinguishing AD from control. Cut off values were determined by maximizing Youden index value. ROC analysis of tau/NPTX2 indicates its diagnostic power is superior to NPTX2 alone, Aβ42, tau or p-tau. AUC: area under ROC curve. Control, n = 61–72; AD, n = 50–58. **p<0.01, ***p<0.001 by two-tailed t test. Data represent mean ± SEM.

The following source data and figure supplements are available for figure 5:

**Source data 1.** Summary of human CSF analysis in Alzheimer's disease (first cohort).

*Figure 5 continued on next page*

Figure 5 continued

**Source data 2.** Summary of human CSF analysis in Alzheimer's disease (second cohort).

**Figure supplement 1.** Reduction of NPTXR levels in lumbar cerebrospinal fluid (CSF) from individuals with AD.

**Figure supplement 2.** Detection of NPTX in human CSF.

**Figure supplement 3.** Development of NPTX2 ELISA assay.

**Figure supplement 4.** Reduction of NPTX levels in second set of CSF from individuals with AD.

**Figure supplement 5.** Positive correlation of NPTXs with CSF tau and p-tau.

## NPTX2 expression correlates with cognitive performance and measures of hippocampal volume

Detailed cognitive data available for the second set of AD subjects revealed positive correlations (p < 0.05) of NPTX2 with the Dementia Rating Scale (DRS) (*Mattis, 1988*), Digital symbol substitution (*Wechsler, 1981*), modified Wisconsin Card Sorting Task (*Nelson, 1976*), Block design Subtest (*Wechsler, 1974*), Visual Reproduction Test [Russell Adaptation WMS; (*Wechsler, 1945*), semantic verbal fluency [SVF (*Borkowski et al., 1967*), California verbal learning test [CVLT (*Delis et al., 1987*) (*Figure 6* and *Table 1*). By contrast, Aß42 levels correlated with SVF and CVLT, while tau and p-tau did not correlate with these behavioral tests (*Table 1*). NPTX2 levels also correlated with hippocampal occupancy (*Heister et al., 2011*), a measure of hippocampal volume, while Aß42, tau and p-tau did not (*Figure 6I and J*, *Table 1*). Measures of hippocampal volume show robust associations with cognitive performance and AD progression (*Fjell et al., 2010*; *Heister et al., 2011*; *Jack et al., 1999*). These findings demonstrate the unique association of CSF NPTX2 expression with cognitive performance and AD pathophysiology.

## Discussion

The present study identifies NPTX2 down-regulation as an important mechanism in AD pathogenesis that is closely linked to cognitive deterioration. The observed reductions of NPTX2 protein in human brain and CSF, and the correlation of CSF NPTX2 with cognitive status and hippocampal volume provide support for the notion that NPTX2 could be an informative biomarker for AD. The potential value of NPTX2 as a biomarker is further advanced by the observation that NPTX2 appears 'orthogonal' to Aß and tau since NPTX2 is not reduced as a direct consequence of Aß amyloid in mouse models, appears independent of Aß in human asymptomatic AD (pre AD), and reductions of CSF NPTX2 do not correlate with reduction of CSF Aß42 or increases of tau/p-tau. Because of the synergistic effects of amyloidosis and NPTX2 down regulation, CSF levels of NPTX2 might distinguish subjects who will be most responsive to amyloid reducing therapy. A recent study reported that CSF NPTX2 may anticipate disease progression (*Swanson et al., 2016*). But the conclusion that NPTX2 is simply an interesting biomarker in AD falls short of the opportunity to implicate a novel mechanism that underlies cognitive failure. In particular, we propose that an important proximal cause of cognitive decline in AD is failure of the adaptive function of pyramidal neurons to modify excitatory drive of fast spiking parvalbumin (PV) interneurons.

The proposed mechanism builds upon prior studies that demonstrate NPTX2 is expressed in pyramidal neurons as an IEG (*Tsui et al., 1996*), and is required for homeostatic scaling of pyramidal neuron excitatory synapses on PV interneurons (*Chang et al., 2010*). One of the remarkable properties of NPTX2 is that it selectively accumulates at excitatory synapses on PV interneurons (*Chang et al., 2010*). NPTX2 is a $Ca^{2+}$ dependent lectin and appears to bind the glycoprotein network that prominently surrounds PV interneurons (*Chang et al., 2010*; *Tsui et al., 1996*). NPTX2 also binds to postsynaptic AMPA receptors (*Lee et al., 2017*; *O'Brien et al., 1999*; *Xu et al., 2003*) and acts to increase local accumulation of AMPA receptors and strengthen synapses. This action of NPTX2 at excitatory synapses on PV interneurons is evident in vivo where genetic deletion of NPTX2

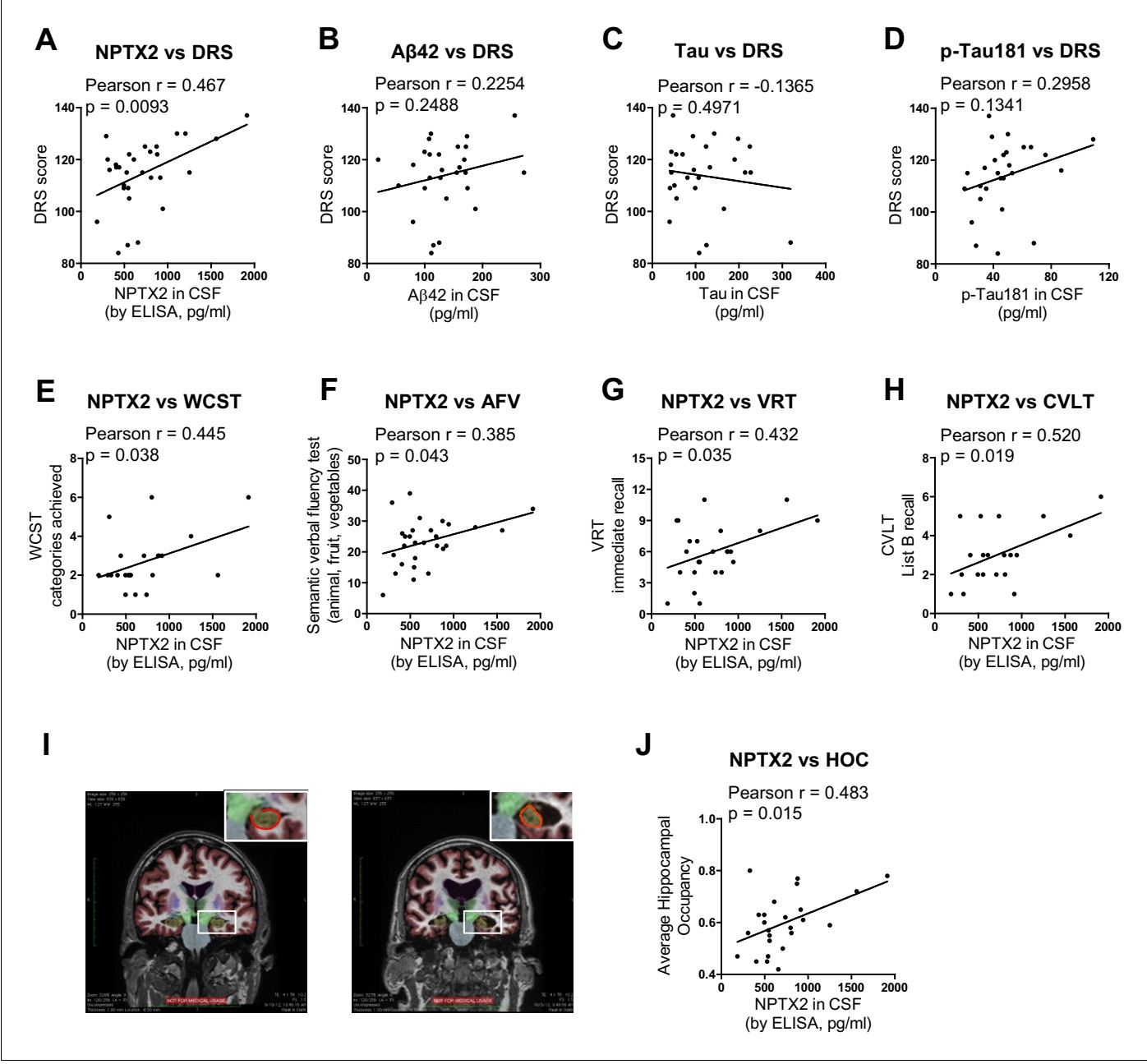

**Figure 6.** NPTX2 expression correlates with cognitive performance and measures of hippocampal volume. (**A**) NPTX2 expression in CSF correlates with cognitive function assayed by DRS in AD group. n = 30. p=0.0093 by Pearson correlation coefficient analysis. DRS: dementia rating scale. (**B–D**) No correlation was observed between CSF Aβ42 (**B**), tau (**C**) or p-tau181 (**D**) with DRS in AD group. n = 28. (**E–H**) NPTX2 expression in CSF correlates with cognitive function assayed by WCST test (**E**), semantic verbal fluency test (**F**), VRT test (**G**) and CVLT test (**H**). n = 20–28. WCST: Wisconsin Card Sorting Task; AFV: Semantic Verbal Fluency Test ('Animals', 'Fruits', 'Vegetables'); VRT: Visual Reproduction Test; CVLT: California Verbal Learning Test. Pearson correlation coefficient analysis was performed. (**I**) Representative images of human brain MRI imaging. (**J**) NPTX2 levels in CSF correlate with hippocampal occupancy. n = 25 AD subjects. HOC: hippocampal occupancy.

results in a selective reduction of pyramidal neuron drive of PV interneurons in developing mouse cortex (*Gu et al., 2013*). *Nptx2*[-/-] alone does not result in marked changes of PV interneuron function as assayed by gamma oscillations in vivo or in acute slices, but *Nptx2*[-/-] combined with *Nptxr*[-/-] resulted in a profound disruption of gamma oscillations and cognitive deficits in combination with a reduction of GluA4 protein in PV-interneurons (*Pelkey et al., 2015*, *2016*). GluA4 protein reduction

**Table 1.** Correlation analysis of CSF biomarkers with hippocampal size and clinical cognitive tests in AD subjects from second cohort.

| | Vs CSF NPTX2 | | | Vs CSF aβ42 | | | Vs CSF tau | | | Vs CSF p-Tau | | |
|---|---|---|---|---|---|---|---|---|---|---|---|---|
| | n | Pearson r | p value | n | Pearson r | p value | n | Pearson r | p value | n | Pearson r | p value |
| Average Hippocampal Occupancy | 25 | 0.483 | 0.015 | 25 | 0.189 | 0.367 | 24 | −0.083 | 0.701 | 24 | 0.360 | 0.084 |
| Dementia Rating Scale | 30 | 0.467 | 0.009 | 28 | 0.225 | 0.249 | 27 | −0.137 | 0.497 | 27 | 0.296 | 0.134 |
| Digit Symbol Substitution | 24 | 0.446 | 0.029 | 23 | 0.127 | 0.565 | 22 | 0.095 | 0.673 | 22 | 0.043 | 0.849 |
| Boston Naming Test | 28 | 0.208 | 0.288 | 27 | 0.098 | 0.628 | 26 | −0.084 | 0.682 | 26 | 0.288 | 0.154 |
| Phonemic Verbal Fluency Test | 28 | 0.200 | 0.308 | 27 | −0.079 | 0.694 | 26 | −0.207 | 0.310 | 26 | 0.133 | 0.519 |
| Semantic Verbal Fluency Test | 28 | 0.385 | 0.043 | 27 | 0.396 | 0.041 | 26 | 0.121 | 0.557 | 26 | 0.036 | 0.863 |
| Wisconsin Card Sorting Task_categories achieved | 22 | 0.445 | 0.038 | 21 | 0.046 | 0.842 | 20 | −0.179 | 0.453 | 20 | −0.057 | 0.812 |
| Wisconsin Card Sorting Task_perseverative errors | 22 | −0.324 | 0.142 | 21 | 0.147 | 0.526 | 20 | 0.013 | 0.956 | 20 | −0.153 | 0.519 |
| Visual Reproduction Test_immediate recall | 24 | 0.432 | 0.035 | 23 | 0.149 | 0.497 | 22 | 0.325 | 0.140 | 22 | 0.292 | 0.187 |
| Visual Reproduction Test_delayed recall | 24 | 0.345 | 0.099 | 23 | 0.123 | 0.577 | 22 | −0.392 | 0.072 | 22 | −0.013 | 0.955 |
| Block Design | 28 | 0.446 | 0.017 | 27 | 0.082 | 0.683 | 26 | −0.111 | 0.590 | 26 | 0.004 | 0.984 |
| Clock Drawing_command | 28 | 0.337 | 0.079 | 27 | 0.040 | 0.845 | 26 | −0.119 | 0.563 | 26 | 0.087 | 0.673 |
| Clock Drawing_copy | 28 | 0.011 | 0.955 | 27 | 0.106 | 0.600 | 26 | −0.246 | 0.225 | 26 | 0.161 | 0.434 |
| California Verbal Learning Test_list B recall | 20 | 0.520 | 0.019 | 20 | 0.668 | 0.001 | 19 | 0.200 | 0.411 | 19 | −0.124 | 0.613 |
| California Verbal Learning Test_recognition | 20 | 0.109 | 0.647 | 20 | −0.096 | 0.687 | 19 | 0.046 | 0.851 | 19 | 0.221 | 0.364 |

Significant correlations (p < 0.05) are highlighted in yellow.

was not associated with a reduced expression of *Gria4* mRNA, TARP protein, parvalbumin immuno-reactivity, or alteration of the perineuronal glycoprotein network. A hypothetical model was suggested in which presynaptic NPTX2 is required to bind postsynaptic GluA4 and prevent receptor turnover in conditions of altered activity consequent to *Nptxr*[-/-] (*Pelkey et al., 2015*, *2016*).

To assess the notion that NPTX2 down-regulation in human brain is indicative of a disruption of the pyramidal neuron-NPTX2-PV interneuron homeostatic mechanism we needed to identify a biomarker of PV-interneuron function that is dependent on NPTX2 expression and that could be quantitatively assayed in human tissue samples. We confirmed in a mouse model that amyloid expression (like *Nptxr*[-/-]) acts in synergy with *Nptx2*[-/-] to down-regulate GluA4 expression and results in anticipated changes of excitability and rhythmicity. These observations validated GluA4 as a biomarker of NPTX2-dependent PV interneuron function in the context of amyloidosis. In human brain we found GluA4 expression was reduced and correlated with NPTX2 expression in the same samples supporting the notion that GluA4 expression is dependent on NPTX2 expression in AD brain. Unexpectedly, GluA4 and NPTX2 correlate within samples in both AD cases and in aged controls. This is notable since in young adults NPTX2 and GluA4 expression do not appear to correlate within samples, and NPTX2 appears higher in young brain than in aged controls. This age-dependence requires further analysis, but suggests that GluA4 expression in PV interneurons becomes dependent on pyramidal neuron expression of NPTX2 during normal aging, and GluA4 expression-PV interneuron function become critically reduced in AD as NPTX2 expression is reduced. Additional studies are also necessary to define the anatomic extent and possible progression of NPTX2 reduction beyond cortical regions.

Both amyloid and NPTX2 result in reduced PV interneuron function and a consequent increase of excitability of pyramidal neurons that are part of the reciprocal circuits with PV interneurons. However, the consequence of amyloid and NPTX2 down-regulation on pyramidal neuron-PV interneuron circuits appear quite different. Amyloid reduces PV interneuron excitability by reducing expression of the voltage sensitive sodium channel Nav1.1 (*Verret et al., 2012*). By contrast, NPTX2 down-regulation reduces pyramidal neuron excitatory drive of PV-interneurons. As an IEG, NPTX2 natural expression is dynamic and limited to neurons that engage in information processing such as place

cells of the hippocampus [see for example the use of IEGs to detect place cells of the hippocampus (*Guzowski et al., 1999*; *Vazdarjanova et al., 2002*). As a result of normal behavioral experience NPTX2 expression establishes 'informational' circuits between pyramidal neurons and PV-interneurons that will preferentially support the future activity of specific ensembles of neurons, for example, during developmental wiring of the visual cortex (*Gu et al., 2013*). Since NPTX2 reduction in AD occurs after accumulation of amyloid and parallels cognitive decline, we hypothesize that circuits can adapt to the effects of amyloid if pyramidal neurons are able to maintain circuit-specific expression of NPTX2. This model suggests that NPTX2 down-regulation is a 'second hit' for pyramidal neuron PV-interneuron circuit dysfunction and cognitive failure in AD. Reciprocally, NPTX2 expression may be considered a resilience factor.

Our model predicts that the combined action of amyloidosis and NPTX2 down-regulation will produce a disruption of the dynamic recruitment of pyramidal neuron-PV interneuron circuits. Consistent with these predictions brain activity is reportedly increased in MCI, early AD and aged controls in regions termed the 'default network' (*Buckner et al., 2005*; *Greicius et al., 2004*; *Sperling et al., 2009*) and dynamic connectivity within and between different resting state networks measured by fMRI blood oxygen level dependent (BOLD) signals is reduced in AD (*Thomas et al., 2014*). Since the capacity for gamma rhythmicity is thought to underlie dynamic connectivity (*Buzsáki and Wang, 2012*; *Engel et al., 2001*), reduced fMRI BOLD signals may be the imaging correlate of dysfunctional pyramidal neuron-PV interneuron circuits. Deficits of gamma oscillations are associated with cognitive impairment in human cognitive disease including AD (*Basar, 2013*). It is further notable that both NPTX2 down-regulation and fMRI functional connectivity similarly correlate with clinical Dementia Rating scores (*Thomas et al., 2014*). Accordingly, we hypothesize that the failure of dynamic circuit properties is most consequential for cognitive dysfunction in AD.

Independent of NPTX2's role in forming 'informational' circuits, increased activity consequent to amyloid and NPTX2 reduction migh accelerate disease progression by placing increased demand on alternative homeostatic mechanisms that weaken pyramidal neuron excitatory synapses, including Arc (*Shepherd et al., 2006*) and Homer1a (*Hu et al., 2010*). Notably, the homeostatic mechanism mediated by Arc generates Aß by enhancing secretase processing of APP (*Wu et al., 2011*). A feed-forward mechanism of synaptic weakening is suggested by the observation that Arc is induced by Aß activation of metabotropic glutamate receptors (*Um et al., 2013*; *Wu et al., 2011*). Arc is reported to drive synapse elimination in the developing cerebellum (*Mikuni et al., 2013*). If Arc's action is increased consequent to amyloid and failure of pyramidal neuron-NPTX2-GluA4 homeostasis it could contribute to increased Aß generation as well as weakening and loss of excitatory synapses on pyramidal neurons. In this model, NPTX2 may be considered a 'resilience factor' that maintains a balance of cell autonomous and circuit-based excitatory homeostasis.

Studies of DS brain suggest that NPTX2 down-regulation may contribute to cognitive deficits in young individuals and disease progression. The availability of CSF and longitudinal follow up will be required to assess this possibility.

Studies directed toward understanding the cause of NPTX2 down-regulation in AD and DS focus attention on post-transcriptional mechanisms that impact mRNA stability. AD and DS share upregulation of miRs that target the same 3'UTR sequence of *Nptx2* mRNA. We hypothesize that NPTX2 is normally regulated by miRs that may mediate the prominent delay in up-regulation of NPTX2 protein [for example after seizure (*O'Brien et al., 1999*), and that this process is dysregulated in AD and DS. It is possible that biochemical pathways accentuated in aging or DS drive pathological miR mechanisms. *miR-96* has been implicated in normal differentiation of the auditory system (*Kuhn et al., 2011*) and cell growth and oncogenesis (*Guttilla and White, 2009*; *Xu et al., 2013*). *miR-182* expression is induced by ß-catenin and in cancer cells contributes to invasiveness by down-regulating RECK, an inhibitor of extracellular metalloproteases (*Chiang et al., 2013*; *Hirata et al., 2013*). *miR-1271*, generated from an intron of *Arl10*, is human specific and targets multiple synaptic genes including GABA synaptic protein gephyrin (*Jensen and Covault, 2011*). ARL10 protein is predicted to play a role in vesicular trafficking similar to ARFs (ADP-ribosylation factors)that are implicated in trafficking synaptic proteins including BACE1 (*Sannerud et al., 2011*) and APOE receptor LRP1 (*Jiang et al., 2014*). It remains a major goal to establish causal mechanisms controlling NPTX2 expression in human brain.

# Materials and methods

## Mouse strains

*Nptx2⁻/⁻* mice in congenic C57BL/6J background were obtained from Mark Perrin's lab. *APPswe/ PS1ΔE9* transgenic mice (*Borchelt et al., 1997*) (here abbreviated *hAPP*) strain was obtained from Dr. Philip Wong. *hAPP* mice with single copy of transgene were crossed with *Nptx2⁻/⁻* to generate *hAPP/Nptx2⁺/⁻*, which were then crossed with *Nptx2⁻/⁻* to generate *hAPP/Nptx2⁻/⁻*. Similarly, WT (C57BL/6J) were crossed to *hAPP* mice to generate *hAPP*/WT, which were crossed to WT to generate cohort. For both WT and *Nptx2* deletion mice cohorts, we confirmed that ~50% of progeny of the final cross carry the *hAPP* transgene, and this assured that mice carry a single copy of the transgene. All procedures involving animals were under the guidelines of JHMI Institutional Animal Care and Use Committee.

## Human specimens

Human brain tissue of Alzheimer's disease (AD) and asymptomatic AD (ASYMAD) was obtained from the Johns Hopkins Brain Resource Center, which includes subjects from the Baltimore Longitudinal Study of Aging. Subjects were recruited by the Clinical Core at Johns Hopkins Alzheimer's Disease Research Center (ADRC) from the community or from the cohort already enrolled in the Baltimore Longitudinal Study on Aging. The assessment procedures have been coordinated by Joint Clinical Core meetings that assure standardization of diagnostic procedures for annual medical, neurologic, psychiatric and neuropsychological evaluations of all subjects. The Neuropathology Core arranges and performs autopsies on clinically well-characterized participants who agreed to autopsy. Results of neuropathological autopsies are then discussed on clinical-pathological conferences attended by members of the Clinical and Neuropathology Cores. Human Down syndrome (DS) brain tissue and control brains that went with those was obtained from two sources – one is University of Maryland brain and tissue bank, and another is Dr. Benjamin Tycko at Taub Institute for Research on Alzheimer's disease and the Aging Brain and Dr. Jerzy Wegiel at Institute for Basic Research in Staten Island. Brain samples were lysed in RIPA buffer plus protease inhibitor cocktail at a dilution factor 1:50 for Western blot analysis.

Human cerebrospinal fluid (CSF) samples were obtained under IRB-approved protocols from participants in the UCSD Alzheimer's Disease Research Center. All participants were given informed consent before taking part in the study. Human CSF samples were dissolved with SDS loading buffer, and 12 μl of CSF were loaded to SDS-PAGE and subsequent Western blot. NPTX2 protein levels in CSF were further quantitated by ELISA assay developed in current study. All CSF samples were frozen at collection and assayed after first thaw. We noted that NPTX protein levels decreased with multiple freeze thaw cycles. To validate the value of NPTX2 as a potential CSF biomarker of AD, two independent cohorts of CSF samples were recruited and analyzed to confirm the finding. Clinic information was blinded during experiment.

## Reagents

Rabbit anti-NPTX1 and anti-NPTX2 were described previously (*Cho et al., 2008*; *Xu et al., 2003*). Mouse anti-NPTX2 monoclonal antibody was made against GST NPTX2 N-terminus (a.a.1–220) fusion protein. Antibody specificity was confirmed with the brain tissue of *NPTX2⁻/⁻* mice. Mouse anti-Arc monoclonal antibody was described previously (*Wu et al., 2011*). All other antibodies are from commercial companies. Sheep anti-NPTXR antibody is from R&D systems (Cat. Number: AF4414; RRID: AB_2153869; Minneapolis, USA); mouse anti-APP N-terminus monoclonal antibody 22C11 is from Millipore (Cat. Number: MAB348; RRID: AB_94882; Massachusetts, USA); rabbit ant-Egr1 (C-19) is from Santa Cruz (Cat. Number: sc-189; RRID: AB_2231020; Dallas, USA); mouse anti-PSD95 is from Thermo Fisher Scientific (Cat. Number: MA1-046; RRID: AB_2092361; Halethorpe, USA); mouse anti-gephyrin is from BD Transduction Laboratories (Cat. Number: 610584; RRID: AB_ 397929, San Jose, USA); mouse anti-Homer1 is from Santa Cruz Biotechnology (Cat. Number: sc-17842; RRID: AB_627742); rabbit anti-GluA4 is from Millipore (Cat. Number: MABN1109); mouse anti-parvalbumin is from Sigma (Cat. Number: P3088; RRID: AB_477329; St. Louis, USA); mouse anti-actin antibody is from Sigma (Cat. Number: A2228; RRID: AB_476697); ECLTM anti-mouse IgG HRP is from GE Healthcare (Cat. Number: NA931V; Halethorpe, USA); ECLTM anti-rabbit IgG HRP is

from GE Healthcare (Cat. Number: NA934V); donkey anti-sheep IgG HRP is from Santa Cruz (Cat. Number: sc-2473; RRID: AB_641190).

Western blot substrate SuperSignal West Pico Luminol Enhancer Solution (Cat. Number: 1859675) and SuperSignal West Pico Stable Peroxide Solution (Cat. Number: 1859674) are from Thermo Fisher Scientific.

## Western Blot

Cultured cells or brain tissue were lysed with a modified RIPA buffer containing 1% Triton, 0.5% Na-deoxycholate, 0.1% SDS, 50 mM NaF, 10 mM $Na_4P_2O_7$, 2 mM $Na_3VO_4$, and protease inhibitor cocktail in PBS, pH 7.4. Protein extracts were separated by 4–12% SDS-PAGE, transferred to PVDF membranes, blocked with 5% non-fat milk, and then probed with primary antibodies for overnight at 4°C. After washes with TBST (TBS with 0.1% Tween-20), membranes were incubated with HRP-conjugated secondary antibodies for 1 hr at room temperature (RT). Immunoreactive bands were visualized by the enhanced chemiluminescent substrate (ECL, Pierce) on X-ray film and quantified using the image software TINA (www.tina-vision.net). Actin and PSD95 were used as loading controls. Proteins migrating similarly in SDS-PAGE gel were assayed on different blots without stripping.

## NPTX2 ELISA

### Preparation of His-tagged NPTX2 standard protein

Full length *Nptx2*-myc in pRK5 vector was used as a template. The N terminal fragment encoding amino acid 1 to 201 was amplified with primers 5' GCAAGGATCCCAAGCCCAGGATAACCC 3' and 5' CATGTCGACTCATGCACTGTTGCCTCTCTC 3', and then cloned into pQE30 vector (Qiagen, Germantown, USA) at sites of BamH1 and SalI. The protein was expressed in XL1-blue host cells induced with 1 mM IPTG. The expressed NPTX2 fragment was purified with NI-NTA agarose column (Qiagen), and the protein concentration was quantified with BCA kit (Thermo Fisher Scientific).

### ELISA assay

Basically, the operation follows the regular process of ELISA. Briefly, 0.5 μg of rabbit anti-NPTX2 antibody in 50 mM $Na_2CO_3$ buffer (pH 9.5) was coated to the ELISA plate (Nunc) at 4°C for overnight. Next day, after treating with blocking solution (5% BSA in PBS) for 1 hr at RT, 100 ul of the series of diluted NPTX2 standard proteins or CSF samples were added into wells and incubated at RT for 1 hr with constant shaking. After washing with TBS, 100 ul of biotinylated mouse anti-NPTX2 antibody was added and incubated at RT for 1 hr. Then, 100 ul of HRP conjugated streptavidin (Biolegend, San Diego, USA) was added and incubated for 1 hr. After washing with TBS, 100 ul of DAB substrate (Biolegend) was applied and incubated for half hour at RT in dark. In the end, 100 ul of 4 M $H_2SO_4$ stopping solution was added and the absorbance was measured at 450 nm. The absolute levels of NPTX2 in CSF were determined by the calculation based on standard curve.

## Immunostaining of GluA4

Human neocortex was fixed by immersion in 10% formaldehyde in PBS (pH 7.4), and then embedded in paraffin and sectioned to 5 μm thickness on slides. Sections were deparaffinized and hydrated by incubating slides at 60°C for 30 min and then transferred into Xylene. The paraffin was removed after 3 changes of Xylene for 5 min. Then, slides were treated 3 min with sequential changes of 100%, 95%, and 70% ethanol and $ddH_2O$ for three times. Protein antigenicity was unmasked by the treatment with 88% formic acid for 5 min. Slides were washed with $ddH_2O$ 3 times for 5 min each. Then, sections were treated with blocking solution (10% horse serum, 0.4% Triton in TBS) for 1 hr at RT and incubated with mouse anti-PV antibody (1:1000, Sigma) and rabbit anti-GluA4 antibody (1:300, Millipore) at 4°C for overnight. After three washes with TBS, tissue sections were incubated with Alexa Fluo-conjugated secondary antibody at RT for 1 hr. Coverslips were mounted on glass slides with ProLong Gold antifade reagent.

## RNA extraction and quantitative PCR

Total RNA and small RNA were extracted by mirVana miRNA isolation kit (Ambion, Thermo Fisher Scientific) according to the manufacturer's protocol. Isolated RNA was treated with DNase to

remove DNA (Turbo DNA-free kit, Ambion). One μg of isolated total RNA was then immediately reverse-transcribed into cDNA using the SuperScript First-Strand Synthesis System for RT-PCR (Invitrogen, Thermo Fisher Scientific). Quantitative PCR was performed with a StepOne Plus machine (Applied Biosystem, Thermo Fisher Scientific) using SYBR green ROX qPCR mastermix in a 96-well optical plate. PCR cycling consists of 95°C for 10 min, followed by 40 cycles of 95°C for 30 s, 64°C for 30 s and 72°C for 30 s. A melt curve was conducted to determine the specificity of PCR amplification. Gapdh was served as an internal control to nomalize data. To assay the direct transcripts pre-mRNA, primers were designed to bind with the intron of genes.

| Assay | Forward primer | Reverse primer |
|---|---|---|
| *Nptx2* pre-mRNA | 5' CTGCCCTCCCGCAAGTATAG 3' | 5' ACCAGAGTGTCCCTACTCCC 3' |
| *Nptx2* mRNA (*Sato et al., 2003*) | 5' CATCGAGCTGCTCATCAAC 3' | 5' CTGCTCTTGTCCAAGGATC 3' |
| *Arl10* pre-mRNA | 5' GTGGGGATGTCGGGAGAAAC 3' | 5' GAAGACCGAAGGCACCAGAG 3' |
| *Arl10* mRNA | 5' AGTTTGTGAGCGAGGTGGAT 3' | 5' CTGGCTGCCAAGAGGAAAAC 3' |
| *Gephyrin* mRNA | 5' CCTCGTCCAGAATACCATCG 3' | 5' CCACGTACTGTTCTGTCTTTGG 3' |
| *Homer1* mRNA | 5' CAAGAACAGAGGGATTCTTTGA 3' | 5' TGCGAAAAGCTTCTTGTTCA 3' |
| *Gapdh* | 5' AGAAGGCTGGGGCTCATTTG 3' | 5' AGGGGCCATCCACAGTCTTC 3' |

For analysis of miRNA abundance, 30 ng of isolated samll RNA was reverse-transcribed using Taqman micoRNA reverse transcription kit (Applied Biosystems), and then subjected to Taqman microRNA assays according to manufacturer's protocol (Applied Biosystems) in StepOne Plus machine. PCR cycling consists of 95°C for 10 min, followed by 40 cycles of 95°C for 15 s, 60°C for 60 s. RNU48 served as an internal control to nomalize data.

## Lentivirus preparation

Precursors of *miR-96*, *miR-1271* and *miR-182* were amplified by PCR from genomic DNA, and inserted into PacI and NheI sites of lentiviral vector pSME2. Constructs were verified by sequencing. The lentiviral production plasmids contain four constructs: the pSME2 lentiviral backbone plasmid and three packaging plasmids pCMV-VSVG, pLP1 and pLP2. All constructs were transformed into stbl3 competent cells (Life Technologies) and then prepared using a standard protocol of Qiagen Maxiprep columns.

Lentivirus was produced and packaged in HEK293T cells. HEK293T cell line was purchased from ATCC (Cat. Number: CRL-3216; RRID: CVCL_0063). No mycoplasma contamination was detected by MycoAlert Mycoplasma Detection Kit (Cat. Number: LT07-218, Lonza, Allendale, USA). HEK293T cells were grown in 175 cm$^2$ flasks that were pre-coated with 0.01% poly L-lysine (PLL) solution and maintained with DMEM containing 10% FBS. To improve transfection efficiency, 25 μM chloroquine was added to the medium of cultured HEK293T cells when culture was 50% confluency, and transfection was performed one hour later. Cells were transfected using FuGene6 (Roche, New York, USA) with the ratio of 1 μg plasmid DNA: 3 μl FuGene6. Eight hours after transfection, 10 μM sodium butyrate was added to the medium to improve transfection efficiency. Culture medium was changed at 24 hr after transfection and collected at 48 hr after medium change (the transfection efficiency was monitored by GFP fluorescence). Virus particles were pelleted by centrifugation at 25,000 rpm for 2 hr at 4°C (Beckman SW 28 rotor). Virus was aliquot and stored at −80°C for future use.

## Neuronal culture

Primary neuronal cultures from embryonic day 17.5 (E17.5) mouse pups were prepared as described previously (*Chang et al., 2010*). Cells were plated on 0.02% PLL-coated wells at a density of 50 × 10$^4$ per well of 12-well plate, and infected with miRNA-expressing lentivirus immediately after plating. Seven days later, cells were harvested for RNA extraction or Western Blot assay.

## DNA methylation assay - pyrosequencing

DNA from postmortem brains of AD patients and healthy control was extracted using QIAamp DNA mini kit (Qiagen), then treated with bisulfite to convert cytosine residues to uracil, but leave

methylated cytosine unaffected (EpiTect bisulfite kit, Qiagen). Bisulfite-treated DNA was amplified by PCR using Qiagen DNA methylation assay PM00126406, then subjected to pyrosequencing to determine the DNA methylation at Johns Hopkins University Genetic Resources Core Facility.

## Luciferase reporter assay

Nptx2 3'UTR was amplified by PCR and inserted downstream of firefly luciferase in a reporter vector pmirGLO (Promega, Madison, USA), which contains another luciferase, renilla luciferase, as an internal reference. Mutation of miR binding site on Nptx2 3'UTR was generated by QuickChange site-directed mutagenesis kit using primers containing mutated nucleotids (Agilent Technologies, Santa Clara, USA). Constructs were verified by sequencing.

HEK293 cells were co-transfected with pmirGLO/ WT or Mut Nptx2 3'UTR and miR mimics or negative control siRNA (Qiagen) by Lipofectamine 2000 (Life Technologies). One day later, cells were lysed and activities of firefly luciferase and renilla luciferase were assayed by Dual-luciferase reporter assay system (Promega). The ratio of firefly luciferase to renilla luciferase activity represented the effects of miR mimics on Nptx2 3'UTR.

## Slice electrophysiology

### Interface electrophysiology experiments

Slices were prepared from wild type, NPTX2$^{-/-}$, hAPP or hAPP/NPTX2$^{-/-}$ mice aged 3 to 4 months. Mice were anaesthetized using isoflurane, decapitated and the brain was immediately immersed in aCSF (at ambient temperature) aerated with 95% $O_2$ and 5% $CO_2$. The aCSF contained (in mM): NMDG (135), KCl (1), $KH_2PO_4$, (1.2), $MgCl_2$ (1.5), $CaCl_2$ (0.5), glucose (10) and choline bicarbonate (20). Horizontal slices (400 µm thick) containing the hippocampus were cut using a Leica VT1000S vibrating microtome (Leica Microsystems, Germany) and were immediately placed into an interface-style chamber (Warner Instruments, CT) containing humidified carbogen gas and perfused with aerated aCSF (1–1.5 ml min$^{-1}$) containing (in mM): NaCl (126), KCl (3), $MgCl_2$ (2), $CaCl_2$ (2), $NaH_2PO_4$ (1.25), glucose (10) and $NaHCO_3$ (26). Slices were incubated for at least one hour before initiating recording. Local field potentials were recorded from CA3 stratum pyramidale using glass pipettes pulled from standard borosilicate glass (3–5 MΩ) and filled with aCSF. Signals were low-pass filtered at 2 kHz and acquired at 10 kHz using an Axon Axopatch 1D amplifier (Molecular Devices, CA), digitized using an Axon Digidata 1322A (Molecular Devices, CA) and captured on a computer running pClamp 9 (Molecular Devices, CA). All Data were imported into Igor Pro 6 (Wavemetrics, OR) using NeuroMatic (ThinkRandom, UK) and analysed using custom-written procedures. Sharp wave ripple oscillation were recorded in drug-free conditions and then gamma oscillations were induced by applying 25 µM carbachol (Sigma-Aldrich, MO) as described (*Fisahn et al., 1998*).

### Data analysis

#### Sharp-wave ripples

SWRs were automatically detected using methods described elsewhere (*Csicsvari et al., 1999*; *Pelkey et al., 2015*), but adapted for in vitro recordings. Extracellular recordings were digitally band-pass filtered between 150 and 250 Hz to reveal SWRs, and the root mean square (RMS) of the filtered recording was used for automatic detection. Periods where the RMS crossed a threshold of 4 SDs above the background were counted as SWR events in our detection algorithm. The nearest peak to the threshold crossing was considered the centre of the SWR and 100 ms on either side of the peak was extracted and the corresponding period of the unfiltered trace was transformed using a Morlet wavelet function. The boundaries of the SWRs were defined as 2 SDs above the baseline power (taken from the first 50 ms at 250 Hz) of the wavelet function. The peak frequency of the SWR was taken from the wavelet transform, and the amplitude calculated by the peak-to-trough amplitude of the band-pass filtered trace. The RMS of the filtered trace was calculated using the SWR boundaries detected from the Wavelet transform, and those events with an average RMS of less than 1.5 SDs above the background were rejected. To prevent contamination of SWR data by single unit activity, all detected events of less than 10 ms duration were automatically rejected by our detection algorithm.

## Gamma oscillations

Power spectra were generated from 400 s of unfiltered recordings during periods of gamma oscillations, and these were normalised by 1/f (after e.g. (*Demanuele et al., 2007*) and the z-score was calculated as before (*Pelkey et al., 2015*). These normalised spectra were used to determine the peak frequency, and power of gamma oscillations. Hypersynchronous bursts / population spikes during gamma oscillations were automatically detected using the root mean square (RMS) of the filtered recording. Traces were digitally high-pass filtered at 300 Hz to reveal single unit activity, and hypersynchronous bursts / population spikes during gamma oscillations were automatically detected using the root mean square (RMS) of the filtered recording. Single units were identified as having amplitude of least three standard deviations above the RMS, and the threshold for detecting hypersynchronous bursts was set at 10 standard deviations above the RMS.

All data were tested for normality and then we used either parametric or nonparametric multiple comparisons tests as appropriate.

## Human behavioral tests (adapted from UCSD ADRC)

All behavioral tests were performed by UCSF ADRC. Summary of tests for which CSF NPTX2 levels showed a correlation of p < 0.05 (*Table 1*).

### Dementia Rating Scale (*Mattis, 1988*)

The DRS is a standardized, 144-point mental status examination with subscales for Attention (37 points), Initiation and Perseveration (37 points), Conceptualization (39 points), Construction (six points), and Memory (25 points). The DRS was administered according to the standardized instructions with the exception that every item was administered to every participant.

### Digit symbol substitution subtest (WAIS-R; [*Wechsler, 1981*])

In this task, subjects are presented with a key, which associates unfamiliar symbols with the numbers 1 through 9. They are then asked to use the key to draw the appropriate symbols below a series of their associated numbers as quickly as possible for 90 s.

### Semantic verbal fluency (*Borkowski et al., 1967*)

The subject is asked to generate orally as many different kinds of animals, fruits, and vegetables as possible within a given time limit. For each category, the subject is allowed one minute to generate items. The subject's score is the total number of items correctly named for the three categories. Perseverations and intrusion errors are also noted. Although fluency tests are sensitive to language dysfunctions and deterioration of semantic knowledge, they can also reflect an individual's capacity to retrieve information from semantic memory. If other language abilities, such as confrontation naming and comprehension are intact, impaired fluency may be due to an inability to initiate systematic retrieval of information in semantic storage (*Butters et al., 1987*).

### Modified Wisconsin Card Sorting Test (WCST) (*Nelson, 1976*)

The Wisconsin Card Sorting Test is widely accepted as a measure of abstracting ability, which is sensitive to frontal lobe pathology. In *Nelson (1976)* modified version of the test, subjects must sort 48 response cards on which are printed one to four symbols (triangle, star, cross, or circle) in one of four colors (red, green, yellow, or blue), according to the proper symbol or color as indicated by a stimulus card. The test continues until six sorts (principles) are achieved or until all 48 response cards have been sorted. The number of sorts achieved, the total number of correct card placements, the number of perseverative errors, and the number of non-perseverative errors is recorded.

### Block design subtest (WISC-R; [*Wechsler, 1974*])

For this constructional task, the subject is presented with four or nine red and white blocks and asked to construct replicas of 11 designs. The blocks are red on two sides, white on two sides, and half white, half red on two sides. The seven four block designs have either 60 or 75 s limits; the three nine block designs, two minute limits. The subject's score depends both upon accuracy and speed. For the first two block designs, the subject copies the examiner's block constructions; for the

remaining nine designs, he copies two-dimensional pictures of the designs. The Block Design Test is sensitive to the early cognitive decline that occurs in either cortical or subcortical dementia.

### Visual reproduction test (Russell Adaptation WMS; [*Wechsler, 1945*])

This test provides a measure of memory for geometric forms. On each of three trials, the subject must reproduce a complex geometric figure from memory immediately following a 10 s study period. Three increasingly complex stimuli containing from 4 to 10 components are presented on successive trials. As a measure of long-term retention, the subjects are asked after 30 min of unrelated testing to again reproduce the figures from memory. A delayed recognition test was also administered, which consists of 18 target items and 18 distractor items. Target items consist of the four complete original designs and 14 partial replications of the four original designs. Finally, the subject is asked to simply copy the stimulus figures in order to assess any visuoperceptual dysfunction that may be contaminating visual memory performance. The subject's reproductions are scored for the number of components present from the original stimulus drawings.

### California verbal learning test (*Delis et al., 1987*)

The CVLT is a list-learning task that assesses multiple cognitive parameters associated with learning and memory, thereby providing an evaluation of the learning process as well as a measure of how much information is acquired and retained. On each of 5 trials, 16 words are presented orally at the rate of one word per second and immediate free recall of the words is elicited. The 16 words consist of four words from each of four semantic categories. Immediately following the fifth presentation/recall trial with the original list, a single presentation/recall trial is presented using a new list (list B) of 16 words. The subject is then asked to again recall the original 16 words, and then to recall the same words when provided with semantic cues. Following a 20 min delay that is filled with unrelated neuropsychological testing, free recall and cued recall of the original 16 word list is again elicited. Finally, a yes/no recognition test is administered consisting of the original 16 words and 28 randomly interspersed distractor words.

Tests for which none of the CSF biomarkers reached p<0.05 included Mini Mental Status Exam (MMSE), Blessed Information-Memory-Concentration Test, Boston Naming Test, Clock Drawing Command, Clock Drawing copy. Tau shows correlation (p=0.012) for CERAD (Consortium to Establish a Registry for Alzheimer's Disease) trial three recall only.

### Mini-Mental state examination (*Folstein et al., 1975*)

The MMSE is a structured scale that briefly assesses orientation to place, orientation to time, registration, attention and concentration, recall, language, and visual construction. It is scored in terms of the number of correctly completed items with lower scores indicative of poorer performance and greater cognitive impairment. The total score ranges from 0 to 30 (perfect performance).

### Phonemic fluency test (*Borkowski et al., 1967*)

On the letter fluency test, the subject is asked to generate orally as many words as possible that begin with the letters 'F,' 'A,' and 'S,' excluding proper names and different forms of the same word. For each letter, the subject is allowed one minute to generate words. Performance is measured by the total number of correct words produced to the three letters. Perseverations (i.e., repetitions of a correct word) and intrusions (i.e., words not beginning with the designated letter) are noted.

### Blessed Information-Memory-Concentration test (*Blessed et al., 1968*)

The test consists of 26 items that assess aspects of orientation (personal information, time, and place), memory (remembering new and remote information), and concentration (counting backwards, saying the months of the year backwards). The IMC test is scored in terms of errors (0 (perfect performance) to 33), with higher scores reflecting poorer performance and greater cognitive impairment.

## Boston naming test (*Kaplan et al., 1983*)

The UCSD abbreviated version of this test requires the subject to name 30 objects depicted in outlined drawings. The drawings are graded in difficulty, with the easiest drawings presented first. If a subject encounters difficulty in naming an object, a stimulus or phonemic cue is provided. The number of spontaneous and cued correct responses, perceptual errors, circumlocutions, paraphasias, and perseverations are used to evaluate the subjects' language performance. In addition to providing quantitative measures of naming ability with and without semantic cueing, the BNT also yields valuable information concerning the processes underlying a given patient's naming errors.

## Clock drawing command and copy tests (*Goodglass and Kaplan, 1972*)

In the first part of this visuoperceptual constructional task, the subject is required to draw a clock with numbers and to set the hands at ten past eleven. In the second part of the test, the subject attempts to copy a picture of a clock with the hands set at ten past eleven. Rouleau and colleagues (*Rouleau et al., 1992*) have developed a 10-point score system for grading an individual's overall performance on the drawing-to-command and copying conditions. A maximum of 2 points is given for the integrity of the clockface; a maximum of 4 points for the presence and sequencing of the numbers; a maximum of 4 points for the presence and placement of the hands.

## CERAD Word-List learning task (*Welsh et al., 1992*)

If it is judged that an overwhelming majority of subjects are able to read, the CERAD Word-List Learning Test will be administered since this test has been widely and effectively used for the assessment of dementia. This task consists of immediate free recall of a 10-item word list assessed over three separate learning trials. The subject is asked to read aloud 10 printed words at a rate of one every two seconds. Immediately following the presentation of the list, the subject is asked to recall as many of the words as possible. Two additional learning trials are immediately administered following the same procedure. After a delay of five to eight minutes filled with unrelated testing, the subject is again asked to recall as many of the words from the 10-item list as he or she can (i.e., delayed recall). A savings score is calculated by dividing the number of words correctly recalled in the delay condition by the number of words recalled on the third learning trial. A delayed word-list recognition task is also administered. In this yes/no recognition task, the subject must identify the original 10 items on a list of these words intermixed with 10 distractor words. As with the Fuld Object Memory Test, the CERAD Word-List Learning Task provides a number of memory measures that are effective for detecting early dementia and for differentiating among different forms of dementia.

## Human brain imaging

Imaging was performed using 3D T1-weighted MRI sequences based on the Alzheimer's Disease Neuroimaging Initiative (ADNI) (*Jack et al., 2008*). For segmentation, NeuroQuant$^{TM}$ a probabilistic-atlas based volumetric MRI software package (CorTechs Labs, Inc, La Jolla, CA), was used to derive regional volumes of the hippocampus and temporal horn using methods previously described (*Brewer et al., 2009*). The hippocampal occupancy score (*Heister et al., 2011*) is a measure of hippocampal volume divided by the sum of hippocampal and temporal horn volumes and is intended to account for regional ex-vacuo expansion of the fluid space adjacent to the hippocampus typically seen when the hippocampus atrophies.

## Receiver Operating Characteristic curve analysis

To evaluate the diagnostic value of CSF biomarkers, receiver operating characteristic (ROC) curve analysis (*Metz, 1978*) was performed. ROC curve was created by plotting the true positive rate (sensitivity) against the false positive rate (100-specificity) for different cut-off points. Each point on the ROC curve represented a sensitivity/specificity pair corresponding to a particular threshold. The area under the ROC curve (AUC) indicated how well two diagnostic groups can be distinguished. Accuracy was defined as the proportion of true results in the whole population [(true positive + true negative)/(true positive + false positive + true negative + false negative)]. For each CSF biomarker, the cut-off point which maximizing Youden index (sensitivity + specificity −1)(*Youden, 1950*) was selected to calculate the accuracy of a CSF biomarker as a diagnostic test.

## Statistical analysis

We used GraphPad PRISM version 5 (RRID: SCR_002798) to perform statistical analyses. Two-tailed t test was employed to analyze difference between two groups. Correlation analysis were performed by Pearson correlation coefficient. Sample sizes of human specimens were estimated by power analysis using G*Power (RRID: SCR_013726).

## Acknowledgements

We wish to thank Drs. Donald Price, Marilyn Albert and Philip Wong and Wayne Silverman for helpful discussions, and Dr. Michael Goggins for kindly providing pancreatic cancer cell DNA.This study was supported by NIMH MH100024 (PFW), (R35 NS-097966) (PFW), P50 AG005146-27 (PFW, JCT), Down Syndrome Research and Treatment Foundation and Research Down Syndrome (MX and RR), NIA AG05131 (DS, SE, DG), Alzheimer's Disease Drug Discovery Foundation (DX, DG) and in part by the Intramural Research Program, National Institute on Aging, and National Institutes on Child Health and Development, NIH.

## Additional information

### Funding

| Funder | Grant reference number | Author |
| --- | --- | --- |
| LuMind Research Down Syndrome Foundation | | Roger H Reeves<br>Paul F Worley<br>Mei-Fang Xiao |
| National Institute on Aging | AG05131 | Douglas Galasko<br>David Salmon<br>Steven Edland |
| National Institute of Mental Health | P50 AG005146-27 | Paul F Worley<br>Juan C Troncoso |
| Alzheimer's drug discovery foundation | 20021106 | Douglas Galasko<br>Paul F Worley |
| National Institute of Mental Health | MH100024 | Paul F Worley |
| National Institute of Mental Health | R35 NS-097966 | Paul F Worley |

The funders had no role in study design, data collection and interpretation, or the decision to submit the work for publication.

### Author contributions

M-FX, Investigation, Methodology, Writing—review and editing, contributes to Conception and design, Acquisition of data, Analysis and interpretation of data, Drafting and revising the article; DX, MTC, KAP, Investigation, Contributes to Conception and design, Acquisition of data, Analysis and interpretation of data; C-CC, YS, Investigation, Contributes to Acquisition of data; JZ, Contributes to Acquisition of data; SR, Contributes to Analysis and interpretation of data; OP, Resources, Contributes to Acquisition of data; DS, Investigation, Methodology, Contributes to Acquisition of data, Analysis and interpretation of data; JB, SE, Investigation, Contributes to Acquisition of data, Analysis and interpretation of data; JW, BT, Resources, Contributes to Conception and design; AS, Investigation, Contributes to Analysis and interpretation of data, Revising the article; RHR, CJM, Supervision, Contributes to Conception and design, Interpretation of data, Revising the article; JCT, DG, Resources, Supervision, Contributes to Conception and design, Interpretation of data, Revising the article; PFW, Conceptualization, Supervision, Writing—original draft, Writing—review and editing, Conception and design, Interpretation of data, Drafting and revising the article

## Author ORCIDs

Michael T Craig, http://orcid.org/0000-0001-8481-6709
Steven Edland, http://orcid.org/0000-0002-1153-7335
Paul F Worley, http://orcid.org/0000-0002-5086-614X

## Ethics

Animal experimentation: This study was performed in strict accordance with the recommendations in the Guide for the Care and Use of Laboratory Animals of the National Institutes of Health. All of the animals were handled according to approved animal care and use committee (ACUC) protocol (Permit Number: MO15M195) of Johns Hopkins University.

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
