## [Decision Letter]

Thank you for submitting your article "NPTX2 and Cognitive Dysfunction in Alzheimer's Disease" for consideration by *eLife*. Your article has been favorably evaluated by Gary Westbrook (Senior Editor) and three reviewers, one of whom is a member of our Board of Reviewing Editors. The reviewers have opted to remain anonymous.

The reviewers have discussed the reviews with one another and the Reviewing Editor has drafted this decision to help you prepare a revised submission.

Summary:

Your manuscript has been evaluated by three reviewers, who provided positive comments on the work. They found the results to be of interest, as the findings imply an influential role of the activity-dependent gene, NPTX2, in Alzheimer's disease. The authors demonstrate that GluA4, an AMPA receptor subunit selectively expressed at ES-PV and essential for inhibitory circuit function that underlies memory, is reduced in Alzheimer's disease cortex. Furthermore, NPTX2 CSF levels are correlated with cognitive performance and hippocampal volume in subjects with AD. Several mechanisms are proposed, including a specific microRNA that targets NPTX2 mRNA and lower GluR4 expression levels, which correlate with NPTX2. The reduction in NPTX2 in the CSF was regarded as a significant marker of the disease.

However, all three reviewers raised a number of questions regarding the reduction and relationship of NPTX2 and GluR4. These issues will need more support and explanation. There were also several experimental issues that should be considered, such as additional EEG measurements. If this is feasible within the two month time frame, it would enhance the analysis. In any case, we believe you will be able to address the majority of the major issues listed below in a revised manuscript.

Essential revisions:

1) NPTX2 CSF levels are correlated with cognitive performance and hippocampal volume in subjects with AD. Thus, NPTX2 might be used a biomarker in the disease. This is potentially important. This reviewer has some reservations regarding the animal studies. In an attempt to provide mechanistic insights on the role of GluA4 and NPTX2 in AD, the authors have investigated a mechanism that might justify the detrimental action of GluA4 and NPTX2 reduction in the disease using animal models. This section of the manuscript is underdeveloped. Specifically, it is not clear to the present reviewer how the reduction in GluA4 and NPTX2 starts and ends up involving the whole brain in addition to PV interneurons. Finally, it is not even clear if pathology involves other parts of the brain.

2) Previous data implied the NPTX2 binds specifically to GluA4 and may lead to clustering and recruitment to excitatory synapses. How does NPTX2 result in a reduction of GluA4? Do these events occur inside neurons? What is the consequence of lower GluR4 levels upon AMPA receptor function?

---

## [Author Response]

Essential revisions:

1) NPTX2 CSF levels are correlated with cognitive performance and hippocampal volume in subjects with AD. Thus, NPTX2 might be used a biomarker in the disease. This is potentially important. This reviewer has some reservations regarding the animal studies. In an attempt to provide mechanistic insights on the role of GluA4 and NPTX2 in AD, the authors have investigated a mechanism that might justify the detrimental action of GluA4 and NPTX2 reduction in the disease using animal models. This section of the manuscript is underdeveloped. Specifically, it is not clear to the present reviewer how the reduction in GluA4 and NPTX2 starts and ends up involving the whole brain in addition to PV interneurons. Finally, it is not even clear if pathology involves other parts of the brain.

We appreciate these comments and agree that the relationship between NPTX2 and GluA4 was underdeveloped in the original manuscript. Because this is an important issue we have revised the manuscript to better explain the rationale and relevance of GluA4 as a biomarker of NPTX2 function in AD brain. The Discussion was rewritten in entirety primarily to address this issue.

In response to the question whether NPTX2 down-regulation involves other parts of the brain and how it may progress, we note this issue in the Discussion as a future direction.

2) Previous data implied the NPTX2 binds specifically to GluA4 and may lead to clustering and recruitment to excitatory synapses. How does NPTX2 result in a reduction of GluA4? Do these events occur inside neurons? What is the consequence of lower GluR4 levels upon AMPA receptor function?

We now know that neuronal pentraxins (NPTX1, NPTX2 and NPTXR) similarly bind to all AMPA receptors. Early work demonstrated that NPTX2 clusters and co-IPs GluA1 and GluA2 subunits (O'Brien et al., 1999) (Xu et al., 2003). In studies examining its direct interaction with AMPA receptors we focused on GluA4 for technical reasons (Sia et al., 2007). NPTX binding to AMPA receptors has been recently reported independently by Dr. Sudhof’s group (Lee et al., 2017).

How does the down-regulation of NPTX2 result in down-regulation of GluA4? As summarized in the revised Discussion, NPTX2 is secreted from pyramidal neurons and acts as a presynaptic factor that clusters postsynaptic GluA4 at the excitatory synapse on PV-interneurons (Chang et al., 2010). The essential contribution of GluA4 to AMPA receptor function in PV-interneurons is rapid channel inactivation (Angulo et al., 1997) (Geiger et al., 1995) (Fuchs et al., 2007). GluA4 is required for effective pyramidal neuron drive of PV interneurons, generation of γ frequency oscillations, and multiple forms of memory including episodic memory (Fuchs et al., 2007).